# Unsupervised discovery of temporal sequences in high-dimensional datasets, with applications to neuroscience

Emily L Mackevicius[1†], Andrew H Bahle[1†], Alex H Williams[2], Shijie Gu[1,3], Natalia I Denisenko[1], Mark S Goldman[4,5*], Michale S Fee[1*]

[1]McGovern Institute for Brain Research, Department of Brain and Cognitive Sciences, Massachusetts Institute of Technology, Cambridge, United States; [2]Neurosciences Program, Stanford University, Stanford, United States; [3]School of Life Sciences and Technology, ShanghaiTech University, Shanghai, China; [4]Center for Neuroscience, Department of Neurobiology, Physiology and Behavior, University of California, Davis, Davis, United States; [5]Department of Ophthamology and Vision Science, University of California, Davis, Davis, United States

**Abstract** Identifying low-dimensional features that describe large-scale neural recordings is a major challenge in neuroscience. Repeated temporal patterns (sequences) are thought to be a salient feature of neural dynamics, but are not succinctly captured by traditional dimensionality reduction techniques. Here, we describe a software toolbox—called seqNMF—with new methods for extracting informative, non-redundant, sequences from high-dimensional neural data, testing the significance of these extracted patterns, and assessing the prevalence of sequential structure in data. We test these methods on simulated data under multiple noise conditions, and on several real neural and behavioral data sets. In hippocampal data, seqNMF identifies neural sequences that match those calculated manually by reference to behavioral events. In songbird data, seqNMF discovers neural sequences in untutored birds that lack stereotyped songs. Thus, by identifying temporal structure directly from neural data, seqNMF enables dissection of complex neural circuits without relying on temporal references from stimuli or behavioral outputs.
DOI: https://doi.org/10.7554/eLife.38471.001

*For correspondence:
msgoldman@ucdavis.edu (MSG);
fee@mit.edu (MSF)

†These authors contributed equally to this work

Competing interests: The authors declare that no competing interests exist.

## Introduction

The ability to detect and analyze temporal sequences embedded in a complex sensory stream is an essential cognitive function, and as such is a necessary capability of neuronal circuits in the brain (*Clegg et al., 1998*; *Janata and Grafton, 2003*; *Bapi et al., 2005*; *Hawkins and Ahmad, 2016*), as well as artificial intelligence systems (*Cui et al., 2016*; *Sutskever et al., 2014*). The detection and characterization of temporal structure in signals is also useful for the analysis of many forms of physical and biological data. In neuroscience, recent advances in technology for electrophysiological and optical measurements of neural activity have enabled the simultaneous recording of hundreds or thousands of neurons (*Chen et al., 2013*; *Kim et al., 2016*; *Scholvin et al., 2016*; *Jun et al., 2017*), in which neuronal dynamics are often structured in sparse sequences (*Hahnloser et al., 2002*; *Harvey et al., 2012*; *MacDonald et al., 2011*; *Okubo et al., 2015*; *Fujisawa et al., 2008*; *Pastalkova et al., 2008*). Such sequences can be identified by averaging across multiple trials, but only in cases where an animal receives a temporally precise sensory stimulus, or executes a sufficiently stereotyped behavioral task.

Neural sequences have been hypothesized to play crucial roles over a much broader range of natural settings, including during learning, sleep, or diseased states (*Mackevicius and Fee, 2018*). In

these applications, it may not be possible to use external timing references, either because behaviors are not stereotyped or are entirely absent. Thus, sequences must be extracted directly from the neuronal data using unsupervised learning methods. Commonly used methods of this type, such as principal component analysis (PCA) or clustering methods, do not efficiently extract sequences, because they typically only model synchronous patterns of activity, rather than extended spatio-temporal motifs of firing.

Existing approaches that search for repeating neural patterns require computationally intensive or statistically challenging analyses (*Brody, 1999*; *Mokeichev et al., 2007*; *Quaglio et al., 2018*; *Brunton et al., 2016*). While progress has been made in analyzing non-synchronous sequential patterns using statistical models that capture cross-correlations between pairs of neurons (*Russo and Durstewitz, 2017*; *Gerstein et al., 2012*; *Schrader et al., 2008*; *Torre et al., 2016*; *Grossberger et al., 2018*; *van der Meij and Voytek, 2018*), such methods may not have statistical power to scale to patterns that include many (more than a few dozen) neurons, may require long periods ($\geq 10^5$ timebins) of stationary data, and may have challenges in dealing with (non-sequential) background activity. For a review highlighting features and limitations of these methods see (*Quaglio et al., 2018*).

Here, we explore a complementary approach, which uses matrix factorization to reconstruct neural dynamics using a small set of exemplar sequences. In particular, we build on convolutional non-negative matrix factorization (convNMF) (*Smaragdis, 2004*; *Smaragdis, 2007*) (*Figure 1B*), which has been previously applied to identify recurring motifs in audio signals such as speech (*O'Grady and Pearlmutter, 2006*; *Smaragdis, 2007*; *Vaz et al., 2016*), as well as neural signals (*Peter et al., 2017*). ConvNMF identifies exemplar patterns (factors) in conjunction with the times and amplitudes of pattern occurrences. This strategy eliminates the need to average activity aligned to any external behavioral references.

While convNMF may produce excellent reconstructions of the data, it does not automatically produce the minimal number of factors required. Indeed, if the number of factors in the convNMF model is greater than the true number of sequences, the algorithm returns overly complex and redundant factorizations. Moreover, in these cases, the sequences extracted by convNMF will often be inconsistent across optimization runs from different initial conditions, complicating scientific interpretations of the results (*Peter et al., 2017*; *Wu et al., 2016*).

To address these concerns, we developed a toolbox of methods, called seqNMF, which includes two different strategies to resolve the problem of redundant factorizations described above. In addition, the toolbox includes methods for promoting potentially desirable features such as orthogonality or sparsity of the spatial and temporal structure of extracted factors, and methods for analyzing the statistical significance and prevalence of the identified sequential structure. To assess these tools, we characterize their performance on synthetic data under a variety of noise conditions and also show that they are able to find sequences in neural data collected from two different animal species using different behavioral protocols and recording technologies. Applied to extracellular recordings from rat hippocampus, seqNMF identifies neural sequences that were previously found by trial-averaging. Applied to functional calcium imaging data recorded in vocal/motor cortex of untutored songbirds, seqNMF robustly identifies neural sequences active in a biologically atypical and overlapping fashion. This finding highlights the utility of our approach to extract sequences without reference to external landmarks; untutored bird songs are so variable that aligning neural activity to song syllables would be difficult and highly subjective.

## Results

### Matrix factorization framework for unsupervised discovery of features in neural data

Matrix factorization underlies many well-known unsupervised learning algorithms, including PCA (*Pearson, 1901*), non-negative matrix factorization (NMF) (*Lee and Seung, 1999*), dictionary learning, and k-means clustering (see *Udell et al., 2016* for a review). We start with a data matrix, $\mathbf{X}$, containing the activity of $N$ neurons at $T$ timepoints. If the neurons exhibit a single repeated pattern of synchronous activity, the entire data matrix can be reconstructed using a column vector $\mathbf{w}$ representing the neural pattern, and a row vector $\mathbf{h}$ representing the times and amplitudes at which that

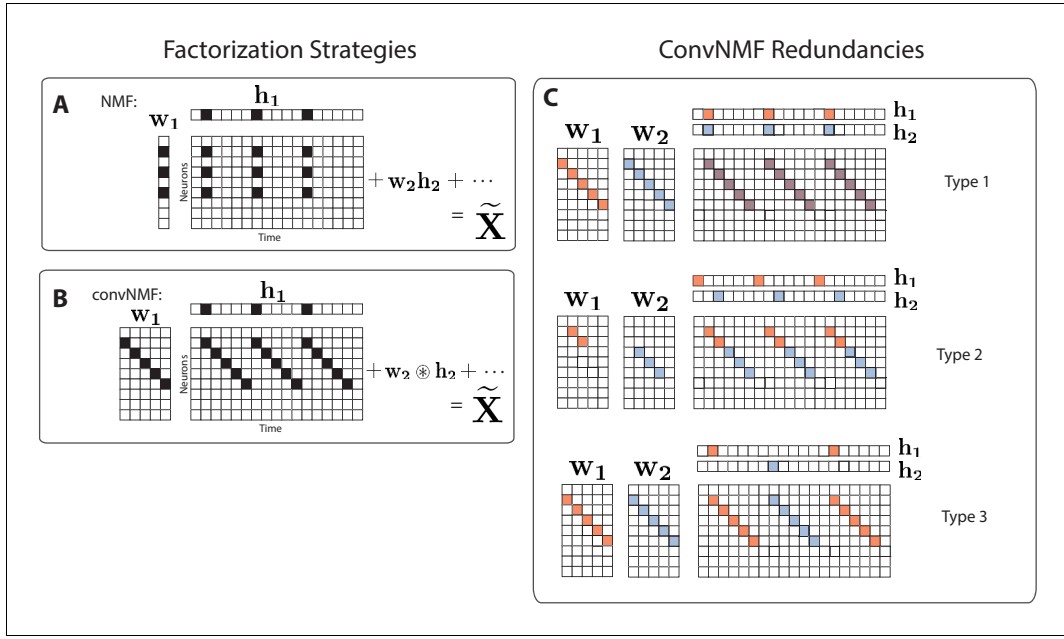

**Figure 1.** Convolutional NMF factorization. (**A**) NMF (non-negative matrix factorization) approximates a data matrix describing the activity of $N$ neurons at $T$ timepoints as a sum of $K$ rank-one matrices. Each matrix is generated as the outer product of two nonnegative vectors: $\mathbf{w}_k$ of length $N$, which stores a neural ensemble, and $\mathbf{h}_k$ of length $T$, which holds the times at which the neural ensemble is active, and the relative amplitudes of this activity. (**B**) Convolutional NMF also approximates an $N \times T$ data matrix as a sum of $K$ matrices. Each matrix is generated as the convolution of two components: a non-negative matrix $\mathbf{w}_k$ of dimension $N \times L$ that stores a sequential pattern of the $N$ neurons at $L$ lags, and a vector of temporal loadings, $\mathbf{h}_k$, which holds the times at which each factor pattern is active in the data, and the relative amplitudes of this activity. (**C**) Three types of inefficiencies present in unregularized convNMF: Type 1, in which two factors are used to reconstruct the same instance of a sequence; Type 2, in which two factors reconstruct a sequence in a piece-wise manner; and Type 3, in which two factors are used to reconstruct different instances of the same sequence. For each case, the factors ($\mathbf{W}$ and $\mathbf{H}$) are shown, as well as the reconstruction ($\widetilde{\mathbf{X}} = \mathbf{W} \circledast \mathbf{H} = \mathbf{w}_1 \circledast \mathbf{h}_1 + \mathbf{w}_2 \circledast \mathbf{h}_2 + \cdots$).

DOI: https://doi.org/10.7554/eLife.38471.002

The following figure supplement is available for figure 1:

**Figure supplement 1.** Quantifying the effect of different penalties on convNMF.

DOI: https://doi.org/10.7554/eLife.38471.003

pattern occurs (temporal loadings). In this case, the data matrix $\mathbf{X}$ is mathematically reconstructed as the outer product of $\mathbf{w}$ and $\mathbf{h}$. If multiple component patterns are present in the data, then each pattern can be reconstructed by a separate outer product, where the reconstructions are summed to approximate the entire data matrix (*Figure 1A*) as follows:

$$\mathbf{X}_{nt} \approx \widetilde{\mathbf{X}}_{nt} = \sum_{k=1}^{K} \mathbf{W}_{nk} \mathbf{H}_{kt} = (\mathbf{WH})_{nt} \tag{1}$$

where $\mathbf{X}_{nt}$ is the $(nt)^{th}$ element of matrix $\mathbf{X}$, that is, the activity of neuron $n$ at time $t$. Here, in order to store $K$ different patterns, $\mathbf{W}$ is a $N \times K$ matrix containing the $K$ exemplar patterns, and $\mathbf{H}$ is a $K \times T$ matrix containing the $K$ timecourses:

$$\mathbf{W} = \begin{bmatrix} | & | & & | \\ \mathbf{w}_1 & \mathbf{w}_2 & \cdots & \mathbf{w}_K \\ | & | & & | \end{bmatrix} \qquad \mathbf{H} = \begin{bmatrix} - & \mathbf{h}_1 & - \\ - & \mathbf{h}_2 & - \\ & \vdots & \\ - & \mathbf{h}_K & - \end{bmatrix} \tag{2}$$

Given a data matrix with unknown patterns, the goal of matrix factorization is to discover a small

set of patterns, $\mathbf{W}$, and a corresponding set of temporal loading vectors, $\mathbf{H}$, that approximate the data. In the case that the number of patterns, $K$, is sufficiently small (less than $N$ and $T$), this corresponds to a dimensionality reduction, whereby the data is expressed in more compact form. PCA additionally requires that the columns of $\mathbf{W}$ and the rows of $\mathbf{H}$ are orthogonal. NMF instead requires that the elements of $\mathbf{W}$ and $\mathbf{H}$ are nonnegative. The discovery of unknown factors is often accomplished by minimizing the following cost function, which measures the element-by-element sum of all squared errors between a reconstruction $\widetilde{\mathbf{X}} = \mathbf{WH}$ and the original data matrix $\mathbf{X}$ using the Frobenius norm, $\|\mathbf{M}\|_F = \sqrt{\sum_{ij}\mathbf{M}_{ij}^2}$:

$$(\mathbf{W}^*, \mathbf{H}^*) = \underset{\mathbf{W},\mathbf{H}}{\arg\min}\|\widetilde{\mathbf{X}} - \mathbf{X}\|_F^2$$

(3)

(Note that other loss functions may be substituted if desired, for example to better reflect the noise statistics; see (*Udell et al., 2016*) for a review). The factors $\mathbf{W}^*$ and $\mathbf{H}^*$ that minimize this cost function produce an optimal reconstruction $\widetilde{\mathbf{X}}^* = \mathbf{W}^*\mathbf{H}^*$. Iterative optimization methods such as gradient descent can be used to search for global minima of the cost function; however, it is often possible for these methods to get caught in local minima. Thus, as described below, it is important to run multiple rounds of optimization to assess the stability/consistency of each model.

While this general strategy works well for extracting synchronous activity, it is unsuitable for discovering temporally extended patterns—first, because each element in a sequence must be represented by a different factor, and second, because NMF assumes that the columns of the data matrix are independent 'samples' of the data, so permutations in time have no effect on the factorization of a given data. It is therefore necessary to adopt a different strategy for temporally extended features.

## Convolutional matrix factorization

Convolutional nonnegative matrix factorization (convNMF) (*Smaragdis, 2004*; *Smaragdis, 2007*) extends NMF to provide a framework for extracting temporal patterns, including sequences, from data. While in classical NMF each factor $\mathbf{W}$ is represented by a single vector (*Figure 1A*), the factors $\mathbf{W}$ in convNMF represent patterns of neural activity over a brief period of time. Each pattern is stored as an $N \times L$ matrix, $\mathbf{w_k}$, where each column (indexed by $\ell = 1$ to $L$) indicates the activity of neurons at different timelags within the pattern (*Figure 1B*). The times at which this pattern/ sequence occurs are encoded in the row vector $\mathbf{h_1}$, as for NMF. The reconstruction is produced by convolving the $N \times L$ pattern with the time series $\mathbf{h_1}$ (*Figure 1B*).

If the data contains multiple patterns, each pattern is captured by a different $N \times L$ matrix and a different associated time series vector $\mathbf{h}$. A collection of $K$ different patterns can be compiled together into an $N \times K \times L$ array (also known as a tensor), $\mathbf{W}$ and a corresponding $K \times T$ time series matrix $\mathbf{H}$. Analogous to NMF, convNMF generates a reconstruction of the data as a sum of $K$ convolutions between each neural activity pattern ($\mathbf{W}$), and its corresponding temporal loadings ($\mathbf{H}$):

$$\mathbf{X}_{nt} \approx \widetilde{\mathbf{X}}_{nt} = \sum_{k=1}^{K}\sum_{\ell=0}^{L-1}\mathbf{W}_{nk\ell}\mathbf{H}_{k(t-\ell)} \equiv (\mathbf{W} \circledast \mathbf{H})_{nt}$$

(4)

The tensor/matrix convolution operator $\circledast$ (notation summary, *Table 1*) reduces to matrix multiplication in the $L = 1$ case, which is equivalent to standard NMF. The quality of this reconstruction can be measured using the same cost function shown in *Equation 3*, and $\mathbf{W}$ and $\mathbf{H}$ may be found iteratively using similar multiplicative gradient descent updates to standard NMF (*Lee and Seung, 1999*; *Smaragdis, 2004*; *Smaragdis, 2007*).

While convNMF can perform extremely well at reconstructing sequential structure, it can be challenging to use when the number of sequences in the data is not known (*Peter et al., 2017*). In this case, a reasonable strategy would be to choose $K$ at least as large as the number of sequences that one might expect in the data. However, if $K$ is greater than the actual number of sequences, convNMF often identifies more significant factors than are minimally required. This is because each sequence in the data may be approximated equally well by a single sequential pattern or by a linear combination of multiple partial patterns. A related problem is that running convNMF from different random initial conditions produces inconsistent results, finding different combinations of partial

**Table 1.** Notation for convolutional matrix factorization

**Shift operator**

The operator $(\overset{\ell\to}{\mathbf{H}})$ shifts a matrix $\mathbf{H}$ in the $\to$ direction by $\ell$ timebins:

$$(\overset{\ell\to}{\mathbf{H}})_{\cdot t} = \mathbf{H}_{\cdot(t-l)} \text{ and likewise } (\overset{\leftarrow\ell}{\mathbf{H}})_{\cdot t} = \mathbf{H}_{\cdot(t+\ell)}$$

where $\cdot$ indicates all elements along the respective matrix dimension.

The shift operator inserts zeros when $(t-\ell) < 0$ or $(t+\ell) > T$

**Tensor convolution operator**

Convolutive matrix factorization reconstructs a data matrix $\mathbf{X}$

using a $N \times K \times L$ tensor $\mathbf{W}$ and a $K \times T$ matrix $\mathbf{H}$:

$$\widetilde{\mathbf{X}} = \mathbf{W} \circledast \mathbf{H} = \sum_\ell \mathbf{W}_{\cdot\cdot\ell} \overset{\ell\to}{\mathbf{H}}$$

Note that each neuron $n$ is reconstructed as the sum of $k$ convolutions:

$$\widetilde{\mathbf{X}}_{nt} = \sum_k \sum_\ell \mathbf{W}_{nk\ell} \mathbf{H}_{k(t-\ell)} \equiv (\mathbf{W} \circledast \mathbf{H})_{nt}$$

**Transpose tensor convolution operator**

The following quantity is useful in several contexts:

$$\mathbf{W} \overset{\top}{\circledast} \mathbf{X} = \sum_\ell (\mathbf{W}_{\cdot\cdot\ell})^\top \overset{\leftarrow\ell}{\mathbf{X}}$$

Note that each element $(\mathbf{W} \overset{\top}{\circledast} \mathbf{X})_{kt} = \sum_l (\mathbf{W}_{\cdot k\ell})^\top \mathbf{X}_{\cdot(t+\ell)} = \sum_l \sum_n \mathbf{W}_{nk\ell} \mathbf{X}_{n(t+\ell)}$ measures

the overlap (correlation) of factor $k$ with the data at time $t$

**convNMF reconstruction**

$$\mathbf{X} \approx \widetilde{\mathbf{X}} = \sum_k \mathbf{W}_{\cdot k \cdot} \circledast \mathbf{H}_{k \cdot} = \mathbf{W} \circledast \mathbf{H}$$

Note that NMF is a special case of convNMF, where $L = 1$

**$L1$ entrywise norm excluding diagonal elements**

For any $K \times K$ matrix $\mathbf{C}$, $\|\mathbf{C}\|_{1,i\neq j} \equiv \sum_k \sum_{j\neq k} \mathbf{C}_{jk}$

**Special matrices**

$\mathbf{1}$ is a $K \times K$ matrix of ones

$\mathbf{I}$ is the $K \times K$ identity matrix

$\mathbf{S}$ is a $T \times T$ smoothing matrix: $\mathbf{S}_{ij} = 1$ when $|i - j| < L$ and otherwise $\mathbf{S}_{ij} = 0$

DOI: https://doi.org/10.7554/eLife.38471.004

patterns on each run (*Peter et al., 2017*). These inconsistency errors fall into three main categories (*Figure 1C*):

- Type 1: Two or more factors are used to reconstruct the same instances of a sequence.
- Type 2: Two or more factors are used to reconstruct temporally different parts of the same sequence, for instance the first half and the second half.
- Type 3: Duplicate factors are used to reconstruct different instances of the same sequence.

Together, these inconsistency errors manifest as strong correlations between different redundant factors, as seen in the similarity of their temporal loadings ($\mathbf{H}$) and/or their exemplar activity patterns ($\mathbf{W}$).

We next describe two strategies for overcoming the redundancy errors described above. Both strategies build on previous work that reduces correlations between factors in NMF. The first strategy is based on regularization, a common technique in optimization that allows the incorporation of constraints or additional information with the goal of improving generalization performance or simplifying solutions to resolve degeneracies (*Hastie et al., 2009*). A second strategy directly estimates the number of underlying sequences by minimizing a measure of correlations between factors (stability NMF; *Wu et al., 2016*).

## Optimization penalties to reduce redundant factors

To reduce the occurrence of redundant factors (and inconsistent factorizations) in convNMF, we sought a principled way of penalizing the correlations between factors by introducing a penalty term, $\mathscr{R}$, into the convNMF cost function:

$$(\mathbf{W}^*, \mathbf{H}^*) = \arg\min_{\mathbf{W}, \mathbf{H}} \left( \|\widetilde{\mathbf{X}} - \mathbf{X}\|_F^2 + \mathscr{R} \right) \tag{5}$$

Regularization has previously been used in NMF to address the problem of duplicated factors, which, similar to Type 1 errors above, present as correlations between the $\mathbf{H}$'s (*Choi, 2008*; *Chen and Cichocki, 2004*). Such correlations are measured by computing the correlation matrix $\mathbf{H}\mathbf{H}^\top$, which contains the correlations between the temporal loadings of every pair of factors. The regularization may be implemented using the penalty term $\mathscr{R} = \lambda \|\mathbf{H}\mathbf{H}^\top\|_{1, i \neq j}$, where the seminorm $\|\cdot\|_{1, i \neq j}$ sums the absolute value of every matrix entry except those along the diagonal (notation summary, *Table 1*) so that correlations between different factors are penalized, while the correlation of each factor with itself is not. Thus, during the minimization process, similar factors compete, and a larger amplitude factor drives down the temporal loading of a correlated smaller factor. The parameter $\lambda$ controls the magnitude of the penalty term $\mathscr{R}$.

In convNMF, a penalty term based on $\mathbf{H}\mathbf{H}^\top$ yields an effective method to prevent errors of Type 1, because it penalizes the associated zero lag correlations. However, it does not prevent errors of the other types, which exhibit different types of correlations. For example, Type 2 errors result in correlated temporal loadings that have a small temporal offset and thus are not detected by $\mathbf{H}\mathbf{H}^\top$. One simple way to address this problem is to smooth the $\mathbf{H}$'s in the penalty term with a square window of length $2L - 1$ using the smoothing matrix $\mathbf{S}$ ($\mathbf{S}_{ij} = 1$ when $|i - j| < L$ and otherwise $\mathbf{S}_{ij} = 0$). The resulting penalty, $\mathscr{R} = \lambda \|\mathbf{H}\mathbf{S}\mathbf{H}^\top\|$, allows factors with small temporal offsets to compete, effectively preventing errors of Types 1 and 2.

This penalty does not prevent errors of Type 3, in which redundant factors with highly similar patterns in $\mathbf{W}$ are used to explain different instances of the same sequence. Such factors have temporal loadings that are segregated in time, and thus have low correlations, to which the cost term $\|\mathbf{H}\mathbf{S}\mathbf{H}^\top\|$ is insensitive. One way to resolve errors of Type 3 might be to include an additional cost term that penalizes the similarity of the factor patterns in $\mathbf{W}$. This has the disadvantage of requiring an extra parameter, namely the $\lambda$ associated with this cost.

Instead we chose an alternative approach to resolve errors of Type 3 that simultaneously detects correlations in $\mathbf{W}$ and $\mathbf{H}$ using a single cross-orthogonality cost term. We note that, for Type 3 errors, redundant $\mathbf{W}$ patterns have a high degree of overlap with the data at the same times, even though their temporal loadings are segregated at different times. To introduce competition between these factors, we first compute, for each pattern in $\mathbf{W}$, its overlap with the data at time $t$. This quantity is captured in symbolic form by $\mathbf{W} \overset{\top}{\circledast} \mathbf{X}$ (see *Table 1*). We then compute the pairwise correlation between the temporal loading of each factor and the overlap of every other factor with the data.

This cross-orthogonality penalty term, which we refer to as 'x-ortho', sums up these correlations across all pairs of factors, implemented as follows:

$$\mathcal{R} = \lambda \| (\mathbf{W} \overset{\top}{\circledast} \mathbf{X}) \mathbf{S} \mathbf{H}^\top \|_{1, i \neq j} \tag{6}$$

When incorporated into the update rules, this causes any factor that has a high overlap with the data to suppress the temporal loadings ($\mathbf{H}$) of any other factors that have high overlap with the data at that time (Further analysis, Appendix 2). Thus, factors compete to explain each feature of the

**Table 2.** Regularized NMF and convNMF: cost functions and algorithms

| NMF | |
|---|---|
| $\mathcal{L} = \frac{1}{2}\|\widetilde{\mathbf{X}} - \mathbf{X}\|_2^2 + \mathcal{R}$ | $\mathbf{W} \leftarrow \mathbf{W} \times \dfrac{\mathbf{X}\mathbf{H}^\top}{\widetilde{\mathbf{X}}\mathbf{H}^\top + \frac{\partial \mathcal{R}}{\partial \mathbf{W}}}$ |
| $\widetilde{\mathbf{X}} = \mathbf{W}\mathbf{H}$ | $\mathbf{H} \leftarrow \mathbf{H} \times \dfrac{\mathbf{W}^\top \mathbf{X}}{\mathbf{W}^\top \widetilde{\mathbf{X}} + \frac{\partial \mathcal{R}}{\partial \mathbf{H}}}$ |
| **convNMF** | |
| $\mathcal{L} = \frac{1}{2}\|\widetilde{\mathbf{X}} - \mathbf{X}\|_2^2 + \mathcal{R}$ | $\mathbf{W}_{\cdot\cdot\ell} \leftarrow \mathbf{W}_{\cdot\cdot\ell} \times \dfrac{\mathbf{X}\overset{\ell\rightarrow}{\mathbf{H}}{}^\top}{\widetilde{\mathbf{X}}\overset{\ell\rightarrow}{\mathbf{H}}{}^\top + \frac{\partial \mathcal{R}}{\partial \mathbf{W}_{\cdot\cdot\ell}}}$ |
| $\widetilde{\mathbf{X}} = \mathbf{W} \circledast \mathbf{H}$ | $\mathbf{H} \leftarrow \mathbf{H} \times \dfrac{\mathbf{W}\overset{\top}{\circledast}\mathbf{X}}{\mathbf{W}\overset{\top}{\circledast}\widetilde{\mathbf{X}} + \frac{\partial \mathcal{R}}{\partial \mathbf{H}}}$ |
| **$L1$ regularization for $\mathbf{H}$ ( $L1$ for $\mathbf{W}$ is analogous)** | |
| $\mathcal{R} = \lambda \|\mathbf{H}\|_1$ | $\frac{\partial \mathcal{R}}{\partial \mathbf{W}_{\cdot\cdot\ell}} = 0$ |
| | $\frac{\partial \mathcal{R}}{\partial \mathbf{H}} = \lambda \mathbf{1}$ |
| **Orthogonality cost for $\mathbf{H}$** | |
| $\mathcal{R} = \frac{\lambda}{2}\|\mathbf{H}\mathbf{H}^\top\|_{1, i \neq j}$ | $\frac{\partial \mathcal{R}}{\partial \mathbf{W}_{\cdot\cdot\ell}} = 0$ |
| | $\frac{\partial \mathcal{R}}{\partial \mathbf{H}} = \lambda(\mathbf{1} - \mathbf{I})\mathbf{H}$ |
| **Smoothed orthogonality cost for $\mathbf{H}$ (favors 'events-based')** | |
| $\mathcal{R} = \frac{\lambda}{2}\|\mathbf{H}\mathbf{S}\mathbf{H}^\top\|_{1, i \neq j}$ | $\frac{\partial \mathcal{R}}{\partial \mathbf{W}_{\cdot\cdot\ell}} = 0$ |
| | $\frac{\partial \mathcal{R}}{\partial \mathbf{H}} = \lambda(\mathbf{1} - \mathbf{I})\mathbf{H}\mathbf{S}$ |
| **Smoothed orthogonality cost for $\mathbf{W}$ (favors 'parts-based')** | |
| $\mathcal{R} = \frac{\lambda}{2}\|\mathbf{W}_{flat}^\top \mathbf{W}_{flat}\|_{1, i \neq j}$ | $\frac{\partial \mathcal{R}}{\partial \mathbf{W}_{\cdot\cdot\ell}} = \lambda \mathbf{W}_{flat}(\mathbf{1} - \mathbf{I})$ |
| where $(\mathbf{W}_{flat})_{nk} = \sum_\ell \mathbf{W}_{nk\ell}$ | $\frac{\partial \mathcal{R}}{\partial \mathbf{H}} = 0$ |
| **Smoothed cross-factor orthogonality (x-ortho penalty)** | |
| $\mathcal{R} = \lambda \|(\mathbf{W}\overset{\top}{\circledast}\mathbf{X})\mathbf{S}\mathbf{H}^\top\|_{1, i \neq j}$ | $\frac{\partial \mathcal{R}}{\partial \mathbf{W}_{\cdot\cdot\ell}} = \lambda \overset{\leftarrow\ell}{\mathbf{X}}\mathbf{S}\mathbf{H}^\top(\mathbf{1} - \mathbf{I})$ |
| | $\frac{\partial \mathcal{R}}{\partial \mathbf{H}} = \lambda(\mathbf{1} - \mathbf{I})\mathbf{W}\overset{\top}{\circledast}\mathbf{X}\mathbf{S}$ |

DOI: https://doi.org/10.7554/eLife.38471.005

data, favoring solutions that use a minimal set of factors to give a good reconstruction. The resulting global cost function is:

$$(\mathbf{W}^*, \mathbf{H}^*) = \underset{\widetilde{\mathbf{W}}, \mathbf{H}}{\arg\min} \left( \|\widetilde{\mathbf{X}} - \mathbf{X}\|_F^2 + \lambda \|(\overset{\top}{\mathbf{W} \circledast \mathbf{X}}) \mathbf{S} \mathbf{H}^\top\|_{1, i \neq j} \right) \tag{7}$$

The update rules for $\mathbf{W}$ and $\mathbf{H}$ are based on the derivatives of this global cost function, leading to a simple modification of the standard multiplicative update rules used for NMF and convNMF (*Lee and Seung, 1999*; *Smaragdis, 2004*; *Smaragdis, 2007*) (*Table 2*). Note that the addition of this cross-orthogonality term does not formally constitute regularization, because it also includes a contribution from the data matrix $\mathbf{X}$, rather than just the model variables $\mathbf{W}$ and $\mathbf{H}$. However, at least for the case that the data is well reconstructed by the sum of all factors, the x-ortho penalty can be shown to be approximated by a formal regularization (Appendix 2). This formal regularization contains both a term corresponding to a weighted smoothed orthogonality penalty on $\mathbf{W}$ and a term corresponding to a weighted smoothed) orthogonality penalty on $\mathbf{H}$, consistent with the observation that the x-ortho penalty simultaneously punishes factor correlations in $\mathbf{W}$ and $\mathbf{H}$.

There is an interesting relation between our method for penalizing correlations and other methods for constraining optimization, namely sparsity. Because of the non-negativity constraint imposed in NMF, correlations can also be reduced by increasing the sparsity of the representation. Previous efforts have been made to minimize redundant factors using sparsity constraints; however, this approach may require penalties on both $\mathbf{W}$ and $\mathbf{H}$, necessitating the selection of two hyper-parameters ($\lambda_w$ and $\lambda_h$) (*Peter et al., 2017*). Since the use of multiple penalty terms increases the complexity of model fitting and selection of parameters, one goal of our work was to design a simple, single penalty function that could regularize both $\mathbf{W}$ and $\mathbf{H}$ simultaneously. The x-ortho penalty described above serves this purpose (*Equation 6*). As we will describe below, the application of sparsity penalties can be very useful for shaping the factors produced by convNMF, and our code includes options for applying sparsity penalties on both $\mathbf{W}$ and $\mathbf{H}$.

## Extracting ground-truth sequences with the x-ortho penalty when the number of sequences is not known

We next examined the effect of the x-ortho penalty on factorizations of sequences in simulated data, with a focus on convergence, consistency of factorizations, the ability of the algorithm to discover the correct number of sequences in the data, and robustness to noise (*Figure 2A*). We first assessed the model's ability to extract three ground-truth sequences lasting 30 timesteps and containing 10 neurons in the absence of noise (*Figure 2A*). The resulting data matrix had a total duration of 15,000 timesteps and contained on average 60±6 instances of each sequence. Neural activation events were represented with an exponential kernel to simulate calcium imaging data. The algorithm was run with the x-ortho penalty for 1000 iterations andit reliably converged to a root-mean-squared-error (RMSE) close to zero (*Figure 2B*). RMSE reached a level within 10% of the asymptotic value in approximately 100 iterations.

While similar RMSE values were achieved using convNMF with and without the x-ortho penalty; the addition of this penalty allowed three ground-truth sequences to be robustly extracted into three separate factors ($\mathbf{w}_1$, $\mathbf{w}_2$, and $\mathbf{w}_3$ in *Figure 2A*) so long as $K$ was chosen to be larger than the true number of sequences. In contrast, convNMF with no penalty converged to inconsistent factorizations from different random initializations when $K$ was chosen to be too large, due to the ambiguities described in *Figure 1*. We quantified the consistency of each model (see Materials and methods), and found that factorizations using the x-ortho penalty demonstrated near perfect consistency across different optimization runs (*Figure 2C*).

We next evaluated the performance of convNMF with and without the x-ortho penalty on datasets with a larger number of sequences. In particular, we set out to observe the effect of the x-ortho penalty on the number of statistically significant factors extracted. Statistical significance was determined based on the overlap of each extracted factor with held out data (see Materials and methods and code package). With the penalty term, the number of significant sequences closely matched the number of ground-truth sequences. Without the penalty, all 20 extracted sequences were significant by our test (*Figure 2D*).

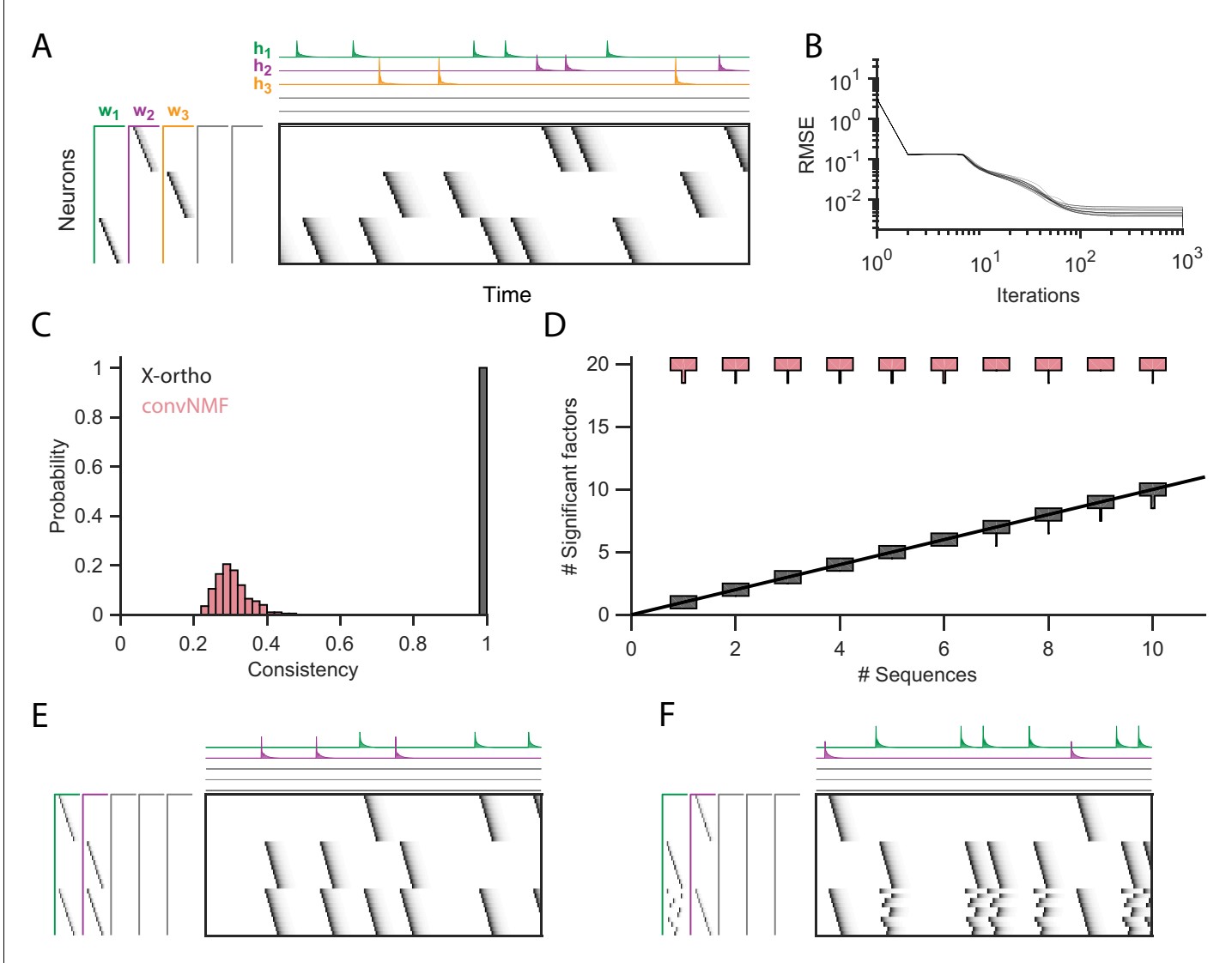

**Figure 2.** Effect of the x-ortho penalty on the factorization of sequences. (**A**) A simulated dataset with three sequences. Also shown is a factorization with x-ortho penalty ($K = 20$, $L = 50$, $\lambda = 0.003$). Each significant factor is shown in a different color. At left are the exemplar patterns (**W**) and on top are the timecourses (**H**). (**B**) Reconstruction error as a function of iteration number. Factorizations were run on a simulated dataset with three sequences and 15,000 timebins ($\approx$ 60 instances of each sequence). Twenty independent runs are shown. Here, the algorithm converges to within 10% of the asymptotic error value within $\approx$ 100 iterations. (**C**) The x-ortho penalty produces more consistent factorizations than unregularized convNMF across 400 independent fits ($K = 20$, $L = 50$, $\lambda = 0.003$). (**D**) The number of statistically significant factors (*Figure 2—figure supplement 1*) vs. the number of ground-truth sequences for factorizations with and without the x-ortho penalty. Shown for each condition is a vertical histogram representing the number of significant factors over 20 runs ($K = 20$, $L = 50$, $\lambda = 0.003$). (**E**) Factorization with x-ortho penalty of two simulated neural sequences with shared neurons that participate at the same latency. (**F**) Same as E but for two simulated neural sequences with shared neurons that participate at different latencies.

DOI: https://doi.org/10.7554/eLife.38471.006

The following figure supplements are available for figure 2:

**Figure supplement 1.** Outline of the procedure used to assess factor significance.
DOI: https://doi.org/10.7554/eLife.38471.007

**Figure supplement 2.** Number of significant factors as a function of $\lambda$ for datasets containing between 1 and 10 sequences.
DOI: https://doi.org/10.7554/eLife.38471.008

We next considered how the x-ortho penalty performs on sequences with more complex structure than the sparse uniform sequences of activity ediscussed above. We further examined the case in which a population of neurons is active in multiple different sequences. Such neurons that are shared across different sequences have been observed in several neuronal datasets (*Okubo et al., 2015*; *Pastalkova et al., 2008*; *Harvey et al., 2012*). For one test, we constructed two sequences in which shared neurons were active at a common pattern of latencies in both sequences; in another test, shared neurons were active in a different pattern of latencies in each sequence. In both tests, factorizations using the x-ortho penalty achieved near-perfect reconstruction error, and consistency was

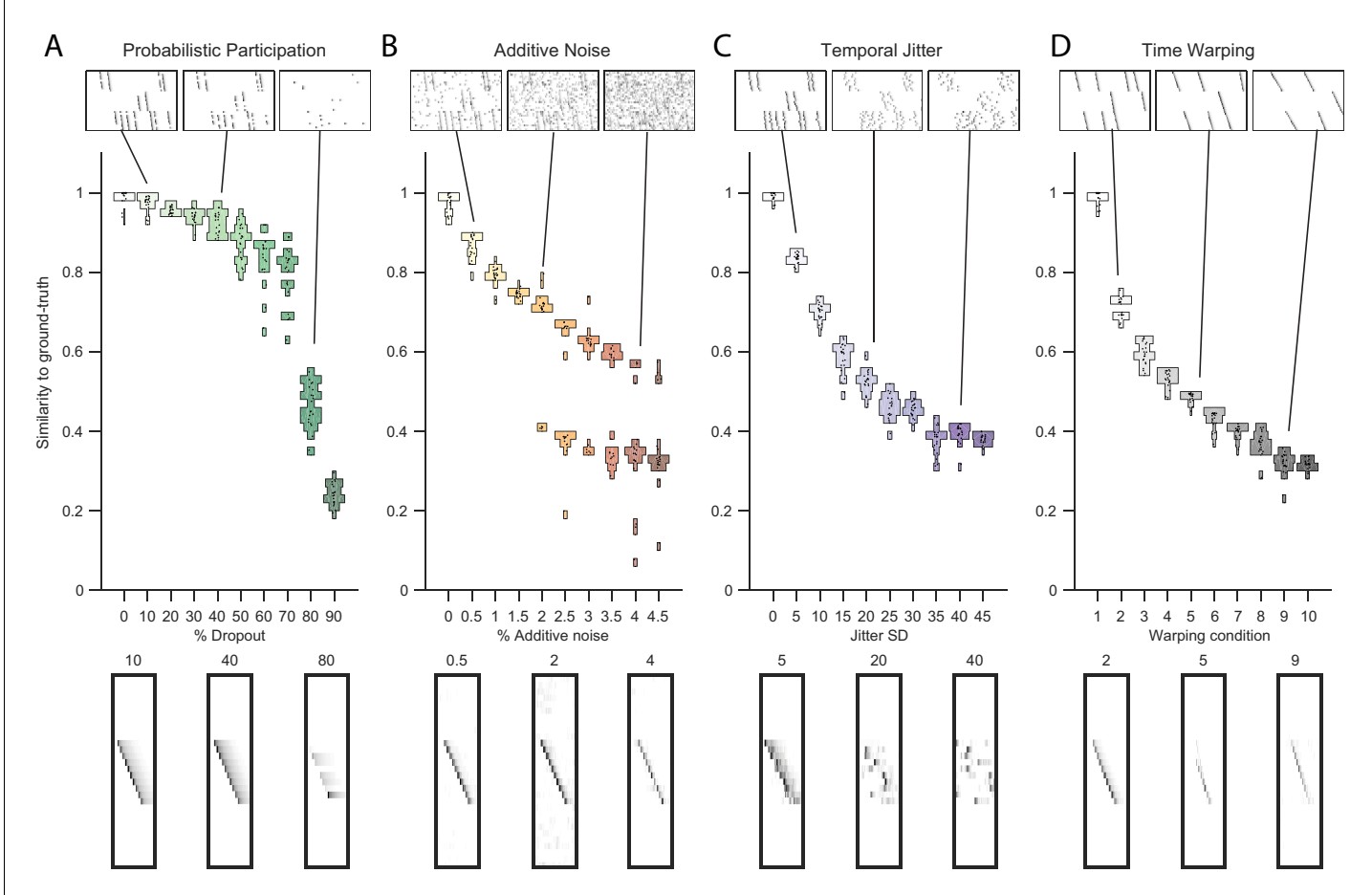

**Figure 3.** Testing factorization performance on sequences contaminated with noise. Performance of the x-ortho penalty was tested under four different noise conditions: (A) probabilistic participation, (B) additive noise, (C) temporal jitter, and (D) sequence warping. For each noise type, we show: (top) examples of synthetic data at three different noise levels; (middle) similarity of extracted factors to ground-truth patterns across a range of noise levels (20 fits for each level); and (bottom) examples of extracted factors **W**'s for one of the ground-truth patterns. Examples are shown at the same three noise levels illustrated in the top row. In these examples, the algorithm was run with $K = 20$, $L = 50$ and $\lambda = 2\lambda_0$ (via the procedure described in *Figure 4*). For C, jitter displacements were draw from a discrete guassian distribution with the standard deviation in timesteps shown above For D, timewarp conditions 1–10 indicate: 0, 66, 133, 200, 266, 333, 400, 466, 533 and 600 max % stretching respectively. For results at different values of $\lambda$, see *Figure 3—figure supplement 1*.

DOI: https://doi.org/10.7554/eLife.38471.009

The following figure supplements are available for figure 3:

**Figure supplement 1.** Robustness to noise at different values of $\lambda$.
DOI: https://doi.org/10.7554/eLife.38471.010

**Figure supplement 2.** Robustness to small dataset size when using the x-ortho penalty.
DOI: https://doi.org/10.7554/eLife.38471.011

**Figure supplement 3.** Robustness to different types of sequences.
DOI: https://doi.org/10.7554/eLife.38471.012

similar to the case with no shared neurons (*Figure 2E,F*). We also examined other types of complex structure and have found that the x-ortho penalty performs well in data with large gaps between activity or with large overlaps of activity between neurons in the sequence. This approach also worked well in cases in which the duration of the activity or the interval between the activity of neurons varied across the sequence (*Figure 3—figure supplement 3*).

## Robustness to noisy data

The cross-orthogonality penalty performed well in the presence of types of noise commonly found in neural data. In particular, we considered: participation noise, in which individual neurons participate probabilistically in instances of a sequence; additive noise, in which neuronal events occur randomly outside of normal sequence patterns; temporal jitter, in which the timing of individual neurons is shifted relative to their typical time in a sequence; and finally, temporal warping, in which each instance of the sequence occurs at a different randomly selected speed. To test the robustness of the algorithm with the x-ortho penalty to each of these noise conditions, we factorized data containing three neural sequences at a variety of noise levels (*Figure 3*, top row). The value of $\lambda$ was chosen using methods described in the next section. Factorizations with the x-ortho penalty proved relatively robust to all four noise types, with a high probability of returning the correct numbers of significant factors (*Figure 4—figure supplement 5*). Furthermore, under low-noise conditions, the algorithm produced factors that were highly similar to ground-truth, and this similarity declined gracefully at higher noise levels (*Figure 3*). Visualization of the extracted factors revealed a good qualitative match to ground-truth sequences even in the presence of high noise except for the case of temporal jitter (*Figure 3*). We also found that the x-ortho penalty allows reliable extraction of sequences in which the duration of each neuron's activity exhibits substantial random variation across different renditions of the sequence, and in which the temporal jitter of neural activity exhibits systematic variation at different points in the sequences (*Figure 3—figure supplement 3*).

Finally, we wondered how our approach with the x-ortho penalty performs on datasets with only a small number of instances of each sequence. We generated data containing different numbers of repetitions ranging from 1 to 20, of each underlying ground-truth sequence. For intermediate levels of additive noise, we found that three repetitions of each sequence were sufficient to correctly extract factors with similarity scores close to those obtained with much larger numbers of repetitions (*Figure 3—figure supplement 2*).

## Methods for choosing an appropriate value of $\lambda$

The x-ortho penalty performs best when the strength of the regularization term (determined by the hyperparameter $\lambda$) is chosen appropriately. For $\lambda$ too small, the behavior of the algorithm approaches that of convNMF, producing a large number of redundant factors with high x-ortho cost. For $\lambda$ too large, all but one of the factors are suppressed to zero amplitude, resulting in a factorization with near-zero x-ortho cost, but with large reconstruction error if multiple sequences are present in the data. Between these extremes, there exists a region in which increasing $\lambda$ produces a rapidly increasing reconstruction error and a rapidly decreasing x-ortho cost. Thus, there is a single point, which we term $\lambda_0$, at which changes in reconstruction cost and changes in x-ortho cost are balanced (*Figure 4A*). We hypothesized that the optimal choice of $\lambda$ (i.e. the one producing the correct number of ground-truth factors) would lie near this point.

To test this intuition, we examined the performance of the x-ortho penalty as a function of $\lambda$ in noisy synthetic data consisting of three non-overlapping sequences (*Figure 4A*). Our analysis revealed that, overall, values of $\lambda$ between $2\lambda_0$ and $5\lambda_0$ performed well for these data across all noise types and levels (*Figure 4B,C*). In general, near-optimal performance was observed over an order of magnitude range of $\lambda$ (*Figure 1*). However, there were systematic variations depending on noise type: for additive noise, performance was better when $\lambda$ was closer to $\lambda_0$, while with other noise types, performance was better at somewhat higher values of $\lambda$s ($\approx 10\lambda_0$).

Similar ranges of $\lambda$ appeared to work for datasets with different numbers of ground-truth sequences—for the datasets used in *Figure 2D*, a range of $\lambda$ between 0.001 and 0.01 returned the correct number of sequences at least 90% of the time for datasets containing between 1 and 10 sequences (*Figure 2—figure supplement 2*). Furthermore, this method for choosing $\lambda$ also worked on datasets containing sequences with shared neurons (*Figure 4—figure supplement 2*).

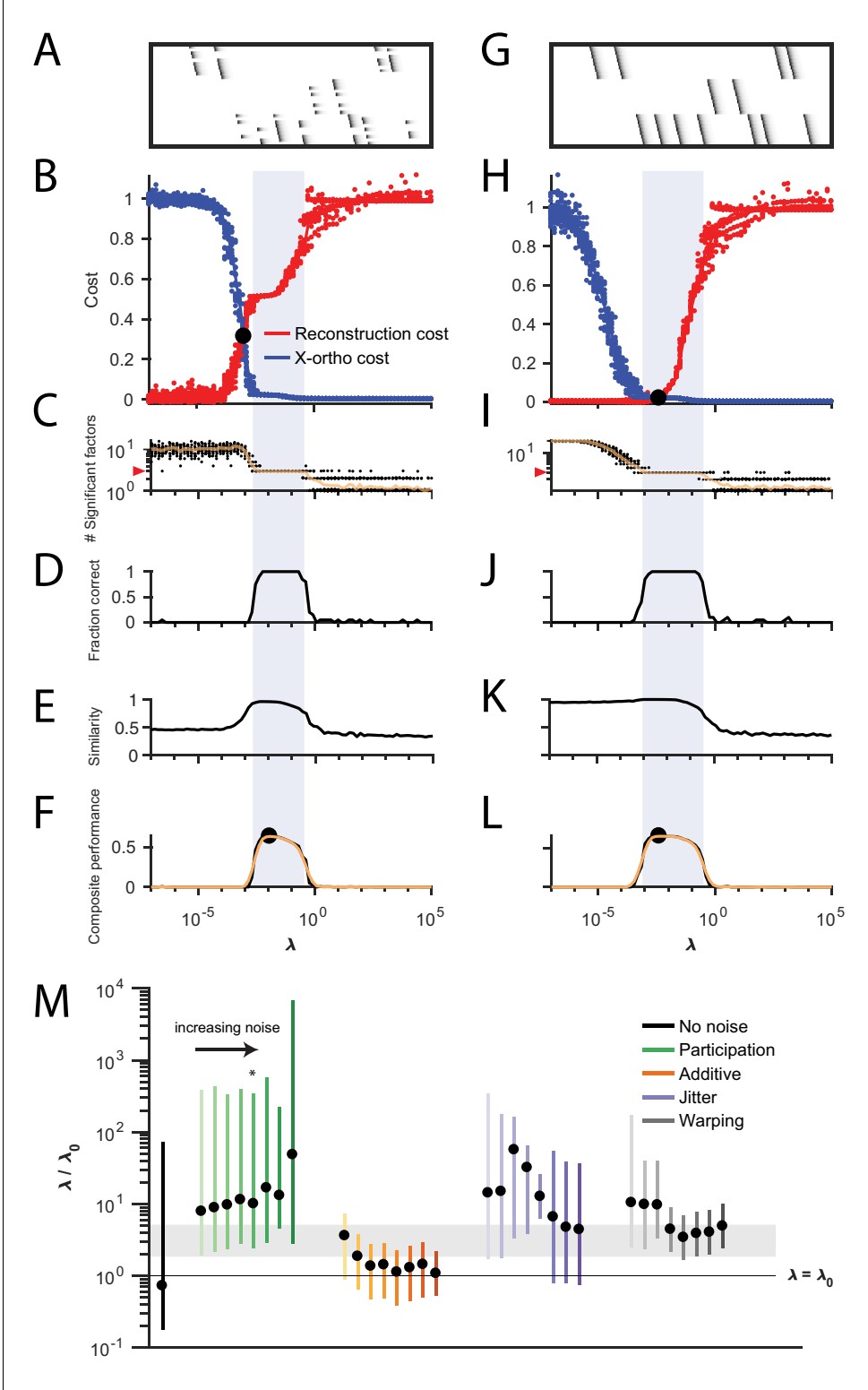

**Figure 4.** Procedure for choosing $\lambda$ for a new dataset based on finding a balance between reconstruction cost and x-ortho cost. (**A**) Simulated data containing three sequences in the presence of participation noise (50% participation probability). This noise condition is used for the tests in (**B–F**). (**B**) Normalized reconstruction cost

($||\widetilde{\mathbf{X}} - \mathbf{X}||_F^2$) and cross-orthogonality cost ($||(\mathbf{W} \overset{\top}{\circledast} \mathbf{X})\mathbf{S}\mathbf{H}^\top||_{1, i \neq j}$) as a function of $\lambda$ for 20 fits of these data. The cross-over point $\lambda_0$ is marked with a black circle. Note that in this plot the reconstruction cost and cross-orthogonality

*Figure 4 continued on next page*

*Figure 4 continued*

cost are normalized to vary between 0 and 1. (**C**) The number of significant factors obtained as a function of $\lambda$; 20 fits, mean plotted in orange. Red arrow at left indicates the correct number of sequences (three). (**D**) Fraction of fits returning the correct number of significant factors as a function of $\lambda$. (**E**) Similarity of extracted factors to ground-truth sequences as a function of $\lambda$. (**F**) Composite performance, as the product of the curves in (**D**) and (**E**) (smoothed using a three sample boxcar, plotted in orange with a circle marking the peak). Shaded region indicates the range of $\lambda$ that works well ($\pm$ half height of composite performance). (**G–L**) same as (**A–F**) but for simulated data containing three noiseless sequences. (**M**) Summary plot showing the range of values of $\lambda$ (vertical bars), relative to the cross-over point $\lambda_0$, that work well for each noise condition ($\pm$ half height points of composite performance). Circles indicate the value of $\lambda$ at the peak of the smoothed composite performance. For each noise type, results for all noise levels from ***Figure 3*** are shown (increasing color saturation at high noise levels; Green, participation: 90, 80, 70, 60, 50, 40, 30, and 20%; Orange, additive noise 0.5, 1, 2, 2.5, 3, 3.5, and 4%; Purple, jitter: SD of the distribution of random jitter: 5, 10, 15, 20, 25, 30, 35, 40, and 45 timesteps; Grey, timewarp: 66, 133, 200, 266, 333, 400, 466, 533, 600, and 666 max % stretching. Asterisk (*) indicates the noise type and level used in panels (**A–F**). Gray band indicates a range between $2\lambda_0$ and $5\lambda_0$, a range that tended to perform well across the different noise conditions. In real data, it may be useful to explore a wider range of $\lambda$.
DOI: https://doi.org/10.7554/eLife.38471.013

The following figure supplements are available for figure 4:

**Figure supplement 1.** Analysis of the best range of $\lambda$.
DOI: https://doi.org/10.7554/eLife.38471.014

**Figure supplement 2.** Procedure for choosing $\lambda$ applied to data with shared neurons.
DOI: https://doi.org/10.7554/eLife.38471.015

**Figure supplement 3.** Using cross-validation on held-out (masked) data to choose $\lambda$.
DOI: https://doi.org/10.7554/eLife.38471.016

**Figure supplement 4.** Quantifying the effect of L1 sparsity penalties on $\mathbf{W}$ and $\mathbf{H}$.
DOI: https://doi.org/10.7554/eLife.38471.017

**Figure supplement 5.** Comparing the performance of convNMF with an x-ortho or a sparsity penalty.
DOI: https://doi.org/10.7554/eLife.38471.018

The value of $\lambda$ may also be determined by cross-validation (see Materials and methods). Indeed, the $\lambda$ chosen with the heuristic described above coincided with a minimum or distinctive feature in the cross-validated test error for all the cases we examined (***Figure 4—figure supplement 3***). The seqNMF code package accompanying this paper provides functions to determine $\lambda$ both by cross-validation or in reference to $\lambda_0$.

## Sparsity constraints to reduce redundant factors

One of the advantages of the x-ortho penalty is that it includes only a single term to penalize correlations between different factors, and thus requires only a single hyperparameter $\lambda$. This contrasts with the approach of incorporating a sparsity constraint on $\mathbf{W}$ and $\mathbf{H}$ of the form $\lambda_w \|\mathbf{W}\|_1 + \lambda_h \|\mathbf{H}\|_1$ (***Peter et al., 2017***). We have found that the performance of the sparsity approach depends on the correct choice of both hyperparameters $\lambda_w$ and $\lambda_h$ (***Figure 4—figure supplement 4***). Given the optimal choice of these parameters, the L1 sparsity constraint yields an overall performance approximately as good as the x-ortho penalty (***Figure 4—figure supplement 4***). However, there are some consistent differences in the performance of the sparsity and x-ortho approaches depending on noise type; an analysis at moderately high noise levels reveals that the x-ortho penalty performs slightly better with warping and participation noise, while the L1 sparsity penalty performs slightly better on data with jitter and additive noise (***Figure 4—figure supplement 5***). However, given the added complexity of choosing two hyperparameters for L1 sparsity, we prefer the x-ortho approach.

## Direct selection of $K$ to reduce redundant factors

An alternative strategy to minimizing redundant factorizations is to estimate the number of underlying sequences and to select the appropriate value of $K$. An approach for choosing the number of factors in regular NMF is to run the algorithm many times with different initial conditions, at different values of $K$, and choose the case with the most consistent and uncorrelated factors. This strategy is called stability NMF (***Wu et al., 2016***) and is similar to other stability-based metrics that have been

used in clustering models (*von Luxburg, 2010*). The stability NMF score, *diss*, is measured between two factorizations, $\mathbf{F}^1 = \{\mathbf{W}^1, \mathbf{H}^1\}$ and $\mathbf{F}^2 = \{\mathbf{W}^2, \mathbf{H}^2\}$, run from different initial conditions:

$$diss(\mathbf{F}^1, \mathbf{F}^2) = \frac{1}{2K}\left(2K - \sum_{j=1}^{K}\max_{1\leq k\leq K}\mathbf{C}_{jk} - \sum_{k=1}^{K}\max_{1\leq j\leq K}\mathbf{C}_{jk}\right)$$

where $\mathbf{C}$ is the cross-correlation matrix between the columns of the matrix $\mathbf{W}^1$ and the the columns of the matrix $\mathbf{W}^2$. Note that *diss* is low when there is a one-to-one mapping between factors in $\mathbf{F}^1$ and $\mathbf{F}^2$, which tends to occur at the correct K in NMF (*Wu et al., 2016*; *Ubaru et al., 2017*). NMF is run many times and the *diss* metric is calculated for all unique pairs. The best value of K is chosen as that which yields the lowest average *diss* metric.

To use this approach for convNMF, we needed to slightly modify the stability NMF *diss* metric. Unlike in NMF, convNMF factors have a temporal degeneracy; that is, one can shift the elements of $\mathbf{h}_k$ by one time step while shifting the elements of $\mathbf{w}_k$ by one step in the opposite direction with little change to the model reconstruction. Thus, rather than computing correlations from the factor patterns $\mathbf{W}$ or loadings $\mathbf{H}$, we computed the *diss* metric using correlations between factor reconstructions ($\widetilde{\mathbf{X}}_k = \mathbf{w_k} \circledast \mathbf{h_k}$).

$$\mathbf{C}_{ij} = \frac{\mathrm{Tr}\left[\widetilde{\mathbf{X}}_i^T\widetilde{\mathbf{X}}_j\right]}{\|\widetilde{\mathbf{X}}_i\|_F\|\widetilde{\mathbf{X}}_j\|_F}$$

where $\mathrm{Tr}[\cdot]$ denotes the trace operator, $\mathrm{Tr}[\mathbf{M}] = \sum_i \mathbf{M}_{ii}$. That is, $\mathbf{C}_{ij}$ measures the correlation between the reconstruction of factor i in $\mathbf{F}^1$ and the reconstruction of factor j in $\mathbf{F}^2$. Here, as for stability NMF, the approach is to run convNMF many times with different numbers of factors (*K*) and choose the *K* which minimizes the *diss* metric.

We evaluated the robustness of this approach in synthetic data with the four noise conditions examined earlier. Synthetic data were constructed with three ground-truth sequences and 20 convNMF factorizations were carried out for each K ranging from 1 to 10. For each K the average *diss* metric was computed over all 20 factorizations. In many cases, the average *diss* metric exhibited a minimum at the ground-truth *K* (*Figure 5—figure supplement 1*). As shown below, this method also appears to be useful for identifying the number of sequences in real neural data.

Not only does the *diss* metric identify factorizations that are highly similar to the ground truth and have the correct number of underlying factors, it also yields factorizations that minimize reconstruction error in held out data (*Figure 5*, *Figure 5—figure supplement 2*), as shown using the same cross-validation procedure described above (*Figure 5—figure supplement 2*). For simulated datasets with participation noise, additive noise, and temporal jitter, there is a clear minimum in the test error at the K given by *diss* metric. In other cases, there is a distinguishing feature such as a kink or a plateau in the test error at this K (*Figure 5—figure supplement 2*).

## Strategies for dealing with ambiguous sequence structure

Some sequences can be interpreted in multiple ways, and these interpretations will correspond to different factorizations. A common example arises when neurons are shared between different sequences, as is shown in *Figure 6A and B*. In this case, there are two ensembles of neurons (1 and 2), that participate in two different types of events. In one event type, ensemble one is active alone, while in the other event type, ensemble one is coactive with ensemble 2. There are two different reasonable factorizations of these data. In one factorization, the two different ensembles are separated into two different factors, while in the other factorization the two different event types are separated into two different factors. We refer to these as 'parts-based' and 'events-based' respectively. Note that these different factorizations may correspond to different intuitions about underlying mechanisms. 'Parts-based' factorizations will be particularly useful for clustering neurons into ensembles, and 'events-based' factorizations will be particularly useful for correlating neural events with behavior.

Here, we show that the addition of penalties on either $\mathbf{W}$ or $\mathbf{H}$ correlations can be used to shape the factorizations of convNMF, with or without the x-ortho penalty, to produce 'parts-based' or 'events-based' factorization. Without this additional control, factorizations may be either 'parts-

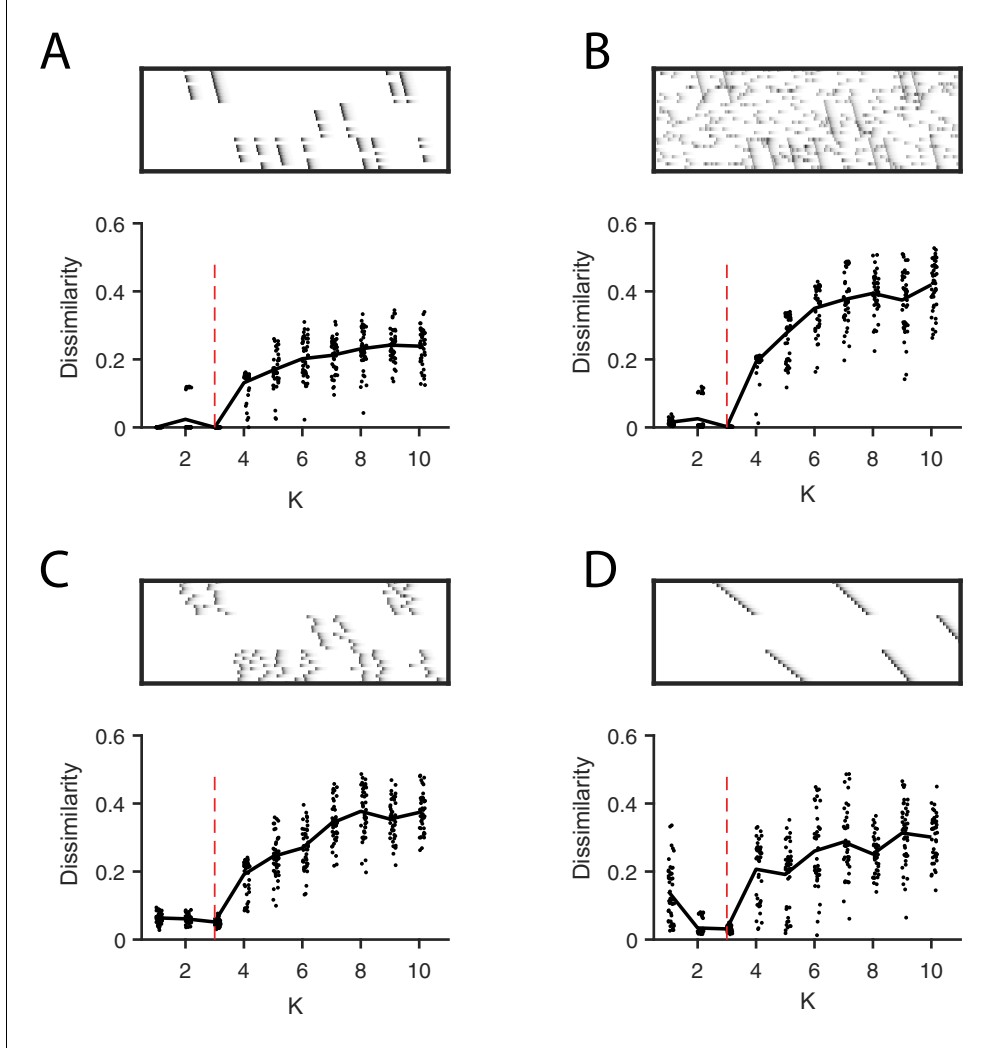

**Figure 5.** Direct selection of K using the *diss* metric, a measure of the dissimilarity between different factorizations. Panels show the distribution of *diss* as a function of K for several different noise conditions. Lower values of *diss* indicate greater consistency or stability of the factorizations, an indication of low factor redundancy. (**A**) probabilistic participation (60%), (**B**) additive noise (2.5% bins), (**C**) timing jitter (SD = 20 bins), and (**D**) sequence warping (max warping = 266%). For each noise type, we show: (top) examples of synthetic data; (bottom) the *diss* metric for 20 fits of convNMF for K from 1 to 10; the black line shows the median of the *diss* metric and the dotted red line shows the true number of factors.

DOI: https://doi.org/10.7554/eLife.38471.019

The following figure supplements are available for figure 5:

**Figure supplement 1.** Direct selection of K using the *diss* metric for all noise conditions.
DOI: https://doi.org/10.7554/eLife.38471.020

**Figure supplement 2.** Estimating the number of sequences in a dataset using cross-validation on randomly masked held-out datapoints.
DOI: https://doi.org/10.7554/eLife.38471.021

based', or 'events-based' depending on initial conditions and the structure of shared neurons activities. This approach works because, in 'events-based' factorization, the $\mathbf{H}$'s are orthogonal (uncorrelated) while the $\mathbf{W}$'s have high overlap; conversely, in the 'parts-based' factorization, the $\mathbf{W}$'s are orthogonal while the $\mathbf{H}$'s are strongly correlated. Note that these correlations in $\mathbf{W}$ or $\mathbf{H}$ are unavoidable in the presence of shared neurons and such correlations do not indicate a redundant factorization. Update rules to implement penalties on correlations in $\mathbf{W}$ or $\mathbf{H}$ are provided in *Table 2*

with derivations in Appendix 1. *Figure 9—figure supplement 2* shows examples of using these penalties on the songbird dataset described in *Figure 9*.

$L1$ regularization is a widely used strategy for achieving sparse model parameters (*Zhang et al., 2016*), and has been incorporated into convNMF in the past (*O'Grady and Pearlmutter, 2006*; *Ramanarayanan et al., 2013*). In some of our datasets, we found it useful to include $L1$ regularization for sparsity. The multiplicative update rules in the presence of $L1$ regularization are included in *Table 2*, and as part of our code package. Sparsity on the matrices $\mathbf{W}$ and $\mathbf{H}$ may be particularly useful in cases when sequences are repeated rhythmically (*Figure 6—figure supplement 1A*). For example, the addition of a sparsity regularizer on the $\mathbf{W}$ update will bias the $\mathbf{W}$ exemplars to include only a single repetition of the repeated sequence, while the addition of a sparsity regularizer on $\mathbf{H}$ will bias the $\mathbf{W}$ exemplars to include multiple repetitions of the repeated sequence. Like the ambiguities described above, these are both valid interpretations of the data, but each may be more useful in different contexts.

## Quantifying the prevalence of sequential structure in a dataset

While sequences may be found in a variety of neural datasets, their importance and prevalence is still a matter of debate and investigation. To address this, we developed a metric to assess how much of the explanatory power of a seqNMF factorization was due to synchronous vs. asynchronous neural firing events. Since convNMF can fit both synchronous and sequential events in a dataset, reconstruction error is not, by itself, diagnostic of the 'sequenciness' of neural activity. Our approach is guided by the observation that in a data matrix with only synchronous temporal structure (i.e. patterns of rank 1), the columns can be permuted without sacrificing convNMF reconstruction error. In contrast, permuting the columns eliminates the ability of convNMF to model data that contains

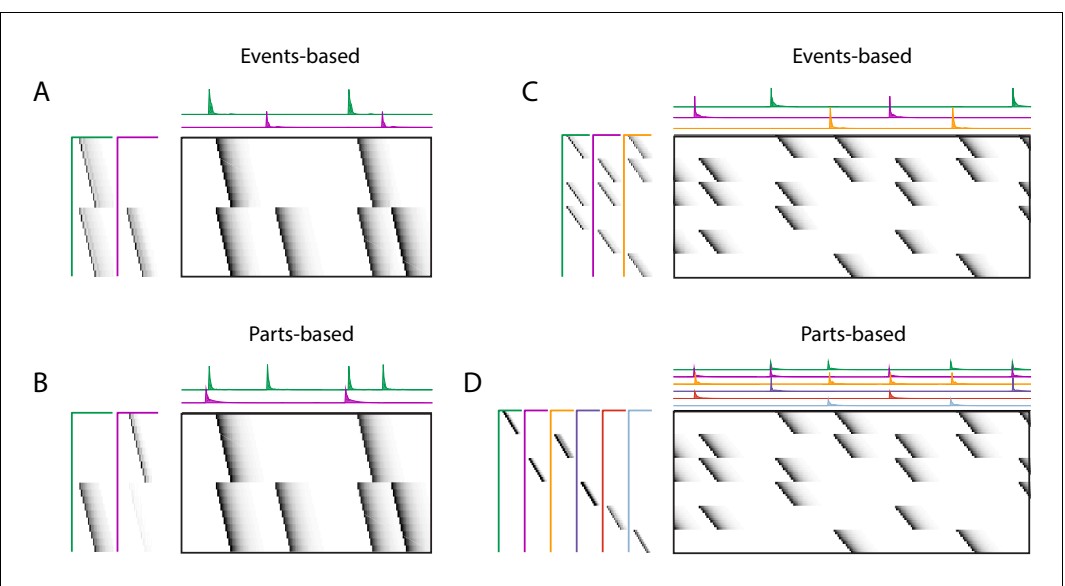

**Figure 6.** Using penalties to bias toward events-based and parts-based factorizations. Datasets that have neurons shared between multiple sequences can be factorized in different ways, emphasizing discrete temporal events (events-based) or component neuronal ensembles (parts-based), by using orthogonality penalties on $\mathbf{H}$ or $\mathbf{W}$ to penalize factor correlations (see *Table 2*). (Left) A dataset with two different ensembles of neurons that participate in two different types of events, with (**A**) events-based factorization obtained using an orthogonality penalty on $\mathbf{H}$ and (**B**) parts-based factorizations obtained using an orthogonality penalty on $\mathbf{W}$. (Right) A dataset with six different ensembles of neurons that participate in three different types of events, with (**C**) events-based and (**D**) parts-based factorizations obtained as in (**A**) and (**B**).

DOI: https://doi.org/10.7554/eLife.38471.022

The following figure supplement is available for figure 6:

**Figure supplement 1.** Biasing factorizations between sparsity in $\mathbf{W}$ or $\mathbf{H}$.

DOI: https://doi.org/10.7554/eLife.38471.023

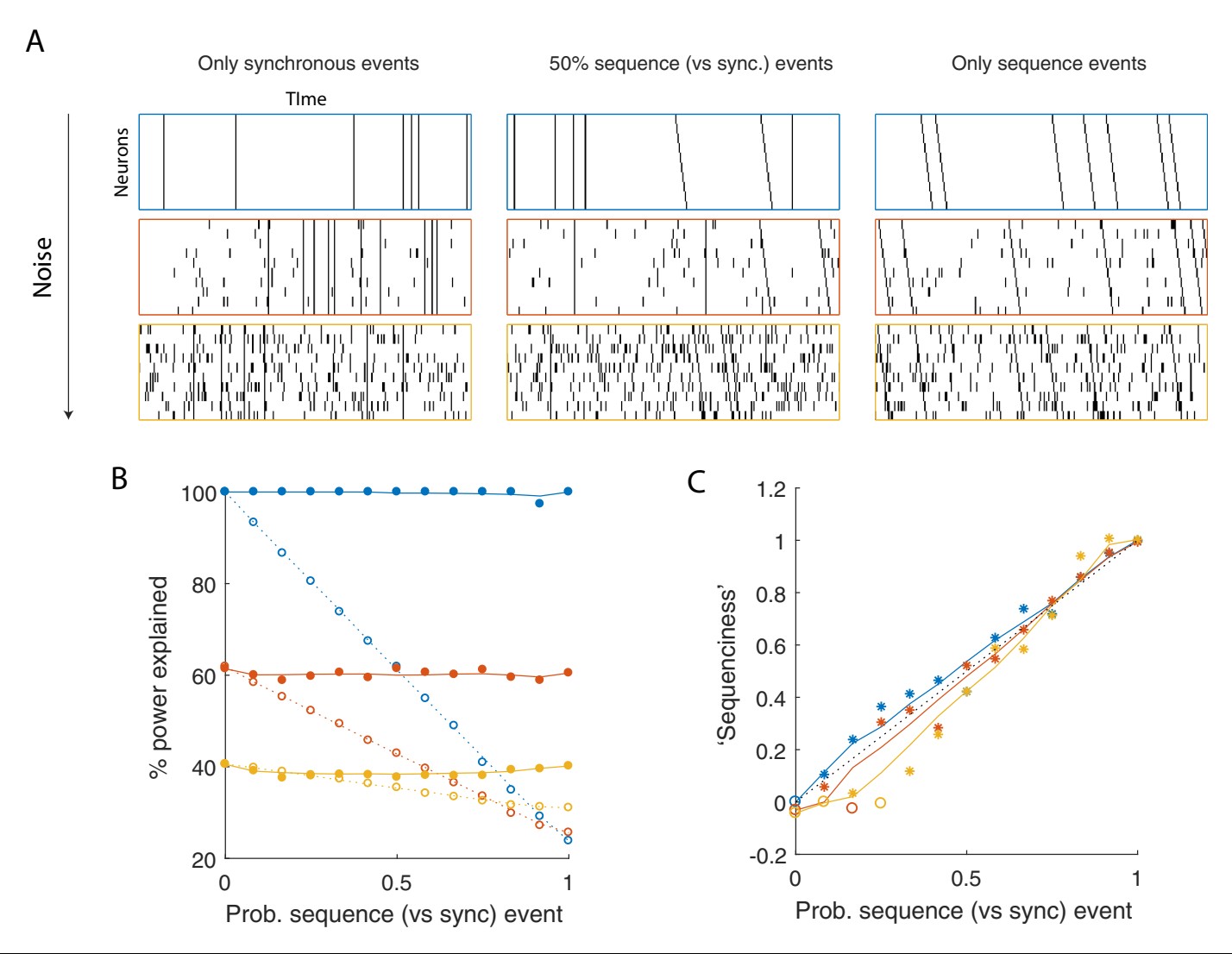

**Figure 7.** Using seqNMF to assess the prevalence of sequences in noisy data. (A) Example simulated datasets. Each dataset contains 10 neurons, with varying amounts of additive noise, and varying proportions of synchronous events versus asynchronous sequences. For the purposes of this figure, 'sequence' refers to a sequential pattern with no synchrony between different neurons in the pattern. The duration of each dataset used below is 3000 times, and here 300 timebins are shown. (B) Median percent power explained by convNMF (L = 12; K = 2; λ=0) for each type of dataset (100 examples of each dataset type). Different colors indicate the three different levels of additive noise shown in A. Solid lines and filled circles indicate results on unshuffled datasets. Note that performance is flat for each noise level, regardless of the probability of sequences vs synchronous events. Dotted lines and open circles indicate results on column-shuffled datasets. When no sequences are present, convNMF performs the same on column-shuffled data. However, when sequences are present, convNMF performs worse on column-shuffled data. (C) For datasets with patterns ranging from exclusively synchronous events to exclusively asynchronous sequences, convNMF was used to generate a 'Sequenciness' score. Colors correspond to different noise levels shown in A. Asterisks denote cases where the power explained exceeds the Bonferroni-corrected significance threshold generated from column-shuffled datasets. Open circles denote cases that do not achieve significance. Note that this significance test is fairly sensitive, detecting even relatively low presence of sequences, and that the 'Sequenciness' score distinguishes between cases where more or less of the dataset consists of sequences.

DOI: https://doi.org/10.7554/eLife.38471.024

sparse temporal sequences (i.e. high rank patterns) but no synchronous structure. We thus compute a 'sequenciness' metric, ranging from 0 to 1, that compares the performance of convNMF on column-shuffled versus non-shuffled data matrices (see Materials and methods), and quantify the performance of this metric in simulated datasets containing synchronous and sequential events with varying prevalence (*Figure 7C*). We found that this metric varies approximately linearly with the

degree to which sequences are present in a dataset. Below, we apply this method to real experimental data and obtain high 'sequenciness' scores, suggesting that convolutional matrix factorization is a well-suited tool for summarizing neural dynamics in these datasets.

## Application of seqNMF to hippocampal sequences

To test the ability of seqNMF to discover patterns in electrophysiological data, we analyzed multi-electrode recordings from rat hippocampus (https://crcns.org/data-sets/hc), which were previously shown to contain sequential patterns of neural firing (*Pastalkova et al., 2015*). Specifically, rats were trained to alternate between left and right turns in a T-maze to earn a water reward. Between alternations, the rats ran on a running wheel during an imposed delay period lasting either 10 or 20 seconds. By averaging spiking activity during the delay period, the authors reported long temporal sequences of neural activity spanning the delay. In some rats, the same sequence occurred on left and right trials, while in other rats, different sequences were active in the delay period during each trial types.

Without reference to the behavioral landmarks, seqNMF was able to extract sequences in both datasets. In Rat 1, seqNMF extracted a single factor, corresponding to a sequence active throughout the running wheel delay period and immediately after, when the rat ran up the stem of the maze (*Figure 8A*); for 10 fits of K ranging from 1 to 10, the average *diss* metric reached a minimum at 1 and with $\lambda = 2\lambda_0$, most runs using the x-ortho penalty extracted a single significant factor (*Figure 8C–E*). Factorizations of thes data with one factor captured 40.8% of the power in the dataset on average, and had a 'sequenciness' score of 0.49. Some runs using the x-ortho penalty extracted two factors (*Figure 8E*), splitting the delay period sequence and the maze stem sequence; this is a reasonable interpretation of the data, and likely results from variability in the relative timing of running wheel and maze stem traversal. At somewhat lower values of $\lambda$, factorizations more often split these sequences into two factors. At even lower values of $\lambda$, factorizations had even more significant factors. Such higher granularity factorizations may correspond to real variants of the sequences, as they generalize to held-out data or may reflect time warping in the data (*Figure 5—figure supplement 2J*). However, a single sequence may be a better description of the data because the *diss* metric displayed a clear minimum at $K = 1$ (*Figure 8C*). In Rat 2, seqNMF typically identified three factors (*Figure 8B*). The first two correspond to distinct sequences active for the duration of the delay period on alternating left and right trials. A third sequence was active immediately following each of the alternating sequences, corresponding to the time at which the animal exits the wheel and runs up the stem of the maze. For 10 fits of K ranging from 1 to 10, the average *diss* metric reached a minimum at three and with $\lambda = 1.5\lambda_0$, most runs with the x-ortho penalty extracted between 2 and 4 factors (*Figure 8F–H*). Factorizations of these data with three factors captured 52.6% of the power in the dataset on average, and had a pattern 'sequenciness' score of 0.85. Taken together, these results suggest that seqNMF can detect multiple neural sequences without the use of behavioral landmarks.

## Application of seqNMF to abnormal sequence development in avian motor cortex

We applied seqNMF methods to analyze functional calcium imaging data recorded in the songbird premotor cortical nucleus HVC during singing. Normal adult birds sing a highly stereotyped song, making it possible to detect sequences by averaging neural activity aligned to the song. Using this approach, it has been shown that HVC neurons generate precisely timed sequences that tile each song syllable (*Hahnloser et al., 2002*; *Picardo et al., 2016*; *Lynch et al., 2016*). Songbirds learn their song by imitation and must hear a tutor to develop normal adult vocalizations. Birds isolated from a tutor sing highly variable and abnormal songs as adults (*Fehér et al., 2009*). Such 'isolate' birds provide an opportunity to study how the absence of normal auditory experience leads to pathological vocal/motor development. However, the high variability of pathological 'isolate' song makes it difficult to identify neural sequences using the standard approach of aligning neural activity to vocal output.

Using seqNMF, we were able to identify repeating neural sequences in isolate songbirds (*Figure 9A*). At the chosen $\lambda$ (*Figure 9B*), x-ortho penalized factorizations typically extracted three significant sequences (*Figure 9C*). Similarly, the *diss* measure has a local minimum at $K = 3$

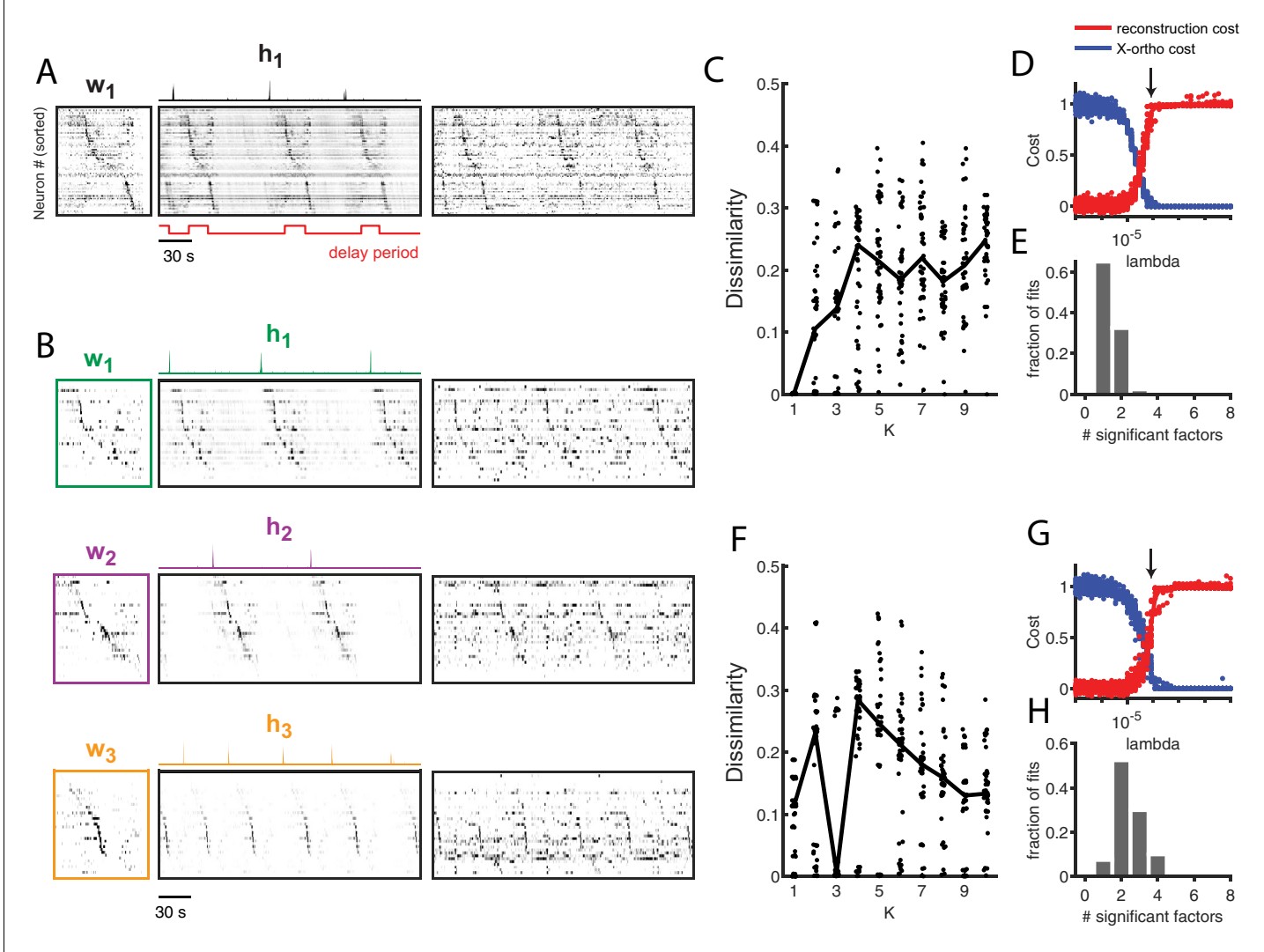

**Figure 8.** Application of seqNMF to extract hippocampal sequences from two rats. (A) Firing rates of 110 neurons recorded in the hippocampus of Rat 1 during an alternating left-right task with a delay period (*Pastalkova et al., 2015*). The single significant extracted x-ortho penalized factor. Both an x-ortho penalized reconstruction of each factor (left) and raw data (right) are shown. Neurons are sorted according to the latency of their peak activation within the factor. The red line shows the onset and offset of the forced delay periods, during which the animal ran on a treadmill. (B) Firing rates of 43 hippocampal neurons recorded in Rat 2 during the same task (*Mizuseki et al., 2013*). Neurons are sorted according to the latency of their peak activation within each of the three significant extracted sequences. The first two factors correspond to left and right trials, and the third corresponds to running along the stem of the maze. (C) The *diss* metric as a function of K for Rat 1. Black line represents the median of the black points. Notice the minimum at K = 1. (D) (Left) Reconstruction (red) and correlation (blue) costs as a function of $\lambda$ for Rat 1. Arrow indicates $\lambda = 8 \times 10^{-5}$, used for the x-ortho penalized factorization shown in (A). (E) Histogram of the number of significant factors across 30 runs of x-ortho penalized convNMF. (D) The *diss* metric as a function of K for Rat 2. Notice the minimum at K = 3. (G–H) Same as in (D–E) but for Rat 2. Arrow indicates $\lambda = 8 \times 10^{-5}$, used for the factorization shown in (B).

DOI: https://doi.org/10.7554/eLife.38471.025

(*Figure 9—figure supplement 1B*). The three-sequence factorization explained 41% of the total power in the dataset, with a sequenciness score of 0.7 andhe extracted sequences included sequences deployed during syllables of abnormally long and variable durations (*Figure 9D–F*, *Figure 9—figure supplement 1A*).

In addition, the extracted sequences exhibit properties not observed in normal adult birds. We see an example of two distinct sequences that sometimes, but not always, co-occur (*Figure 9*). We observe that a shorter sequence (green) occurs alone on some syllable renditions while a second,

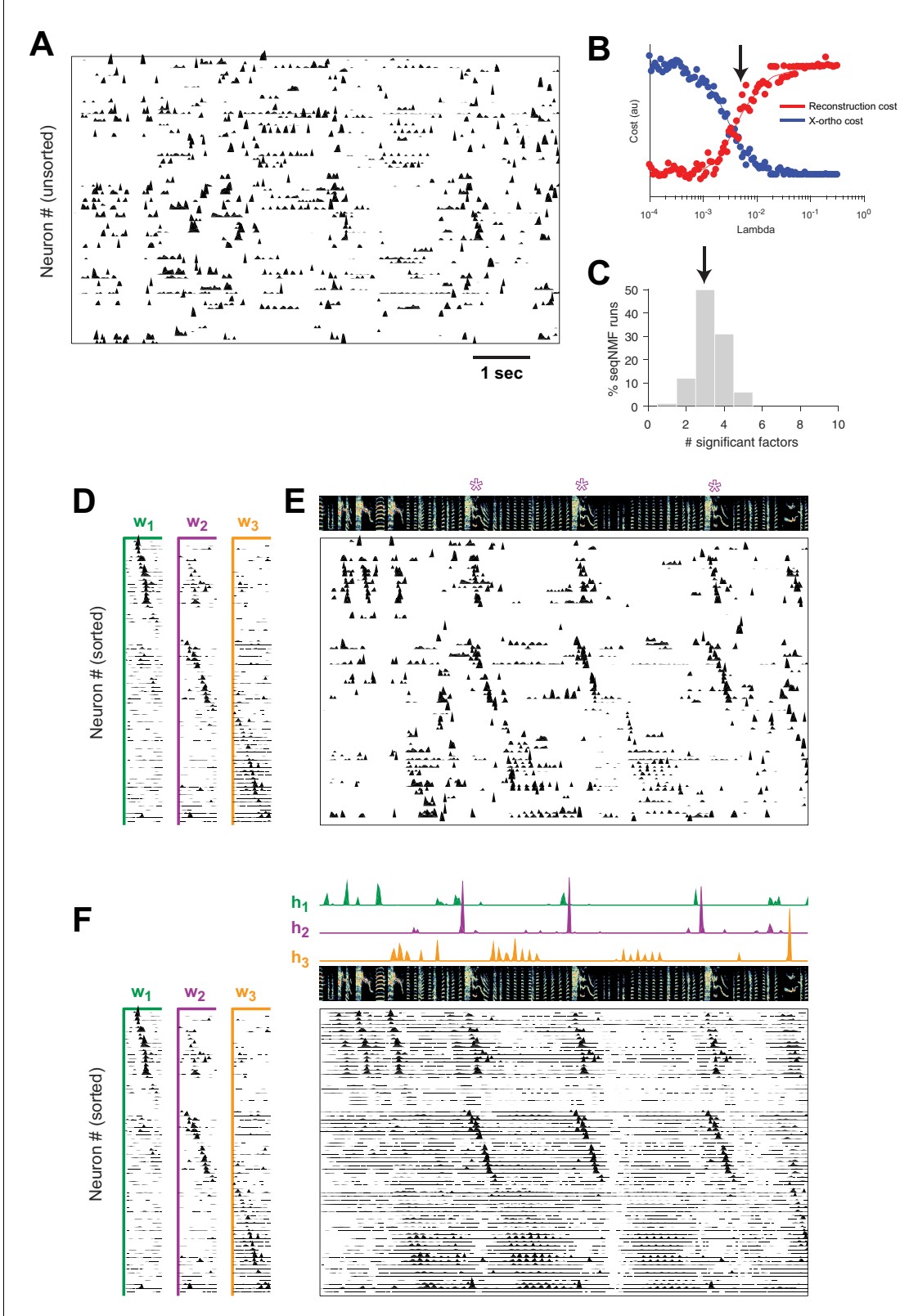

**Figure 9.** SeqNMF applied to calcium imaging data from a singing isolate bird reveals abnormal sequence deployment. (**A**) Functional calcium signals recorded from 75 neurons, unsorted, in a singing isolate bird. (**B**) Reconstruction and cross-orthogonality cost as a function of $\lambda$. The arrow at $\lambda = 0.005$ indicates the value selected for the rest of the analysis. (**C**) Number of significant factors for 100 runs with the x-ortho penalty with $K = 10$, $\lambda = 0.005$. Arrow indicates three is the most common number of significant factors. (**D**) X-ortho factor exemplars (**W**'s). Neurons are grouped according to the

*Figure 9 continued on next page*

*Figure 9 continued*

factor in which they have peak activation, and within each group neurons are sorted by the latency of their peak activation within the factor. (E) The same data shown in (A), after sorting neurons by their latency within each factor as in (D). A spectrogram of the bird's song is shown at top, with a purple '*' denoting syllable variants correlated with $\mathbf{w}_2$. (F) Same as (E), but showing reconstructed data rather than calcium signals. Shown at top are the temporal loadings (**H**) of each factor.

DOI: https://doi.org/10.7554/eLife.38471.026

The following figure supplements are available for figure 9:

**Figure supplement 1.** Further analysis of sequences.

DOI: https://doi.org/10.7554/eLife.38471.027

**Figure supplement 2.** Events-based and parts-based factorizations of songbird data.

DOI: https://doi.org/10.7554/eLife.38471.028

---

longer sequence (purple) occurs simultaneously on other syllable renditions. We found that biasing x-ortho penalized convNMF towards 'parts-based' or 'events-based' factorizations gives a useful tool to visualize this feature of the data (*Figure 9—figure supplement 2*). This probabilistic overlap of different sequences is highly atypical in normal adult birds (*Hahnloser et al., 2002*; *Long et al., 2010*; *Picardo et al., 2016*; *Lynch et al., 2016*) and is associated with abnormal variations in syllable structure—in this case resulting in a longer variant of the syllable when both sequences co-occur. This acoustic variation is a characteristic pathology of isolate song (*Fehér et al., 2009*).

Thus, even though we observe HVC generating sequences in the absence of a tutor, it appears that these sequences are deployed in a highly abnormal fashion.

## Application of seqNMF to a behavioral dataset: song spectrograms

Although we have focused on the application of seqNMF to neural activity data, these methods naturally extend to other types of high-dimensional datasets, including behavioral data with applications to neuroscience. The neural mechanisms underlying song production and learning in songbirds is an area of active research. However, the identification and labeling of song syllables in acoustic recordings is challenging, particularly in young birds in which song syllables are highly variable. Because automatic segmentation and clustering often fail, song syllables are still routinely labelled by hand (*Okubo et al., 2015*). We tested whether seqNMF, applied to a spectrographic representation of zebra finch vocalizations, is able to extract meaningful features in behavioral data. Using the x-ortho penalty, factorizations correctly identified repeated acoustic patterns in juvenile songs, placing each distinct syllable type into a different factor (*Figure 10*). The resulting classifications agree with previously published hand-labeled syllable types (*Okubo et al., 2015*). A similar approach could be applied to other behavioral data, for example movement data or human speech, and could facilitate the study of neural mechanisms underlying even earlier and more variable stages of learning. Indeed, convNMF was originally developed for application to spectrograms (*Smaragdis, 2004*); notably it has been suggested that auditory cortex may use similar computations to represent and parse natural statistics (*Młynarski and McDermott, 2018*).

## Discussion

As neuroscientists strive to record larger datasets, there is a need for rigorous tools to reveal underlying structure in high-dimensional data (*Gao and Ganguli, 2015*; *Sejnowski et al., 2014*; *Churchland and Abbott, 2016*; *Bzdok and Yeo, 2017*). In particular, sequential structure is increasingly regarded as a fundamental property of neuronal circuits (*Hahnloser et al., 2002*; *Harvey et al., 2012*; *Okubo et al., 2015*; *Pastalkova et al., 2008*), but standardized statistical approaches for extracting such structure have not been widely adopted or agreed upon. Extracting sequences is particularly challenging when animal behaviors are variable (e.g. during learning) or absent entirely (e.g. during sleep).

Here, we explored a simple matrix factorization-based approach to identify neural sequences without reference to animal behavior. The convNMF model elegantly captures sequential structure in an unsupervised manner (*Smaragdis, 2004*; *Smaragdis, 2007*; *Peter et al., 2017*). However, in datasets where the number of sequences is not known, convNMF may return inefficient and inconsistent factorizations. To address these challenges, we introduced a new regularization term to penalize

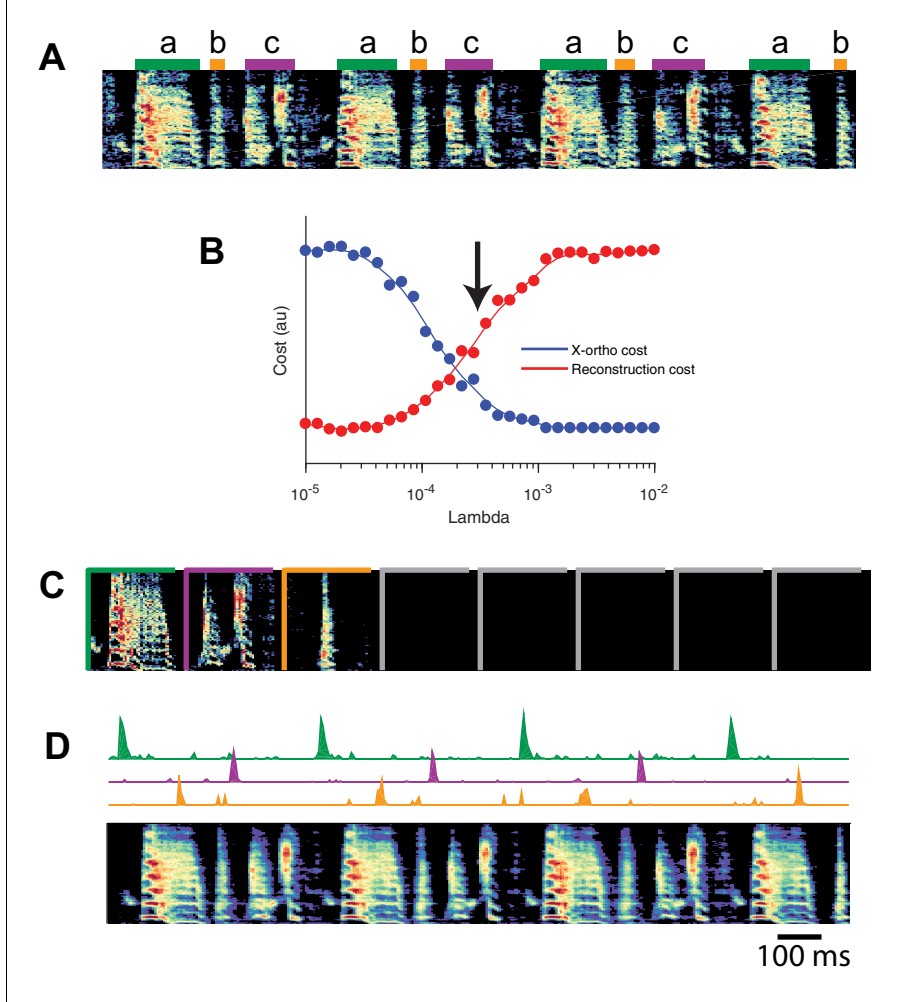

**Figure 10.** SeqNMF applied to song spectrograms. (A) Spectrogram of juvenile song, with hand-labeled syllable types (*Okubo et al., 2015*). (B) Reconstruction cost and x-ortho cost for these data as a function of $\lambda$. Arrow denotes $\lambda = 0.0003$, which was used to run convNMF with the x-ortho penalty (C) $\mathbf{W}$'s for this song, fit with $K = 8$, $L = 200ms$, $\lambda = 0.0003$. Note that there are three non-empty factors, corresponding to the three hand-labeled syllables a, b, and c. (D) X-ortho penalized $\mathbf{H}$'s (for the three non-empty factors) and reconstruction of the song shown in (A) using these factors.
DOI: https://doi.org/10.7554/eLife.38471.029

correlated factorizations, and developed a new dissimilarity measure to assess model stability. Both proposed methods can be used to infer the number of sequences in neural data and are highly robust to noise. For example, even when (synthetic) neurons participate probabilistically in sequences at a rate of 50%, the model typically identifies factors with greater than 80% similarity to the ground truth (*Figure 3A*). Additionally, these methods perform well even with very limited amounts of data: for example successfully extracting sequences that only appear a handful of times in a noisy data stream (*Figure 3—figure supplement 2*).

The x-ortho penalty developed in this paper may represent a useful improvement over traditional orthogonality regularizations or suggest how traditional regularization penalties may be usefully modified. First, it simultaneously provides a penalty on correlations in both $\mathbf{W}$ and $\mathbf{H}$, thus simplifying analyses by having only one penalty term. Second, although the x-ortho penalty does not formally constitute regularization due to its inclusion of the data $\mathbf{X}$, we have described how the penalty can be approximated by a data-free regularization with potentially useful properties (Appendix 2). Specifically, the data-free regularization contains terms corresponding to weighted orthogonality in

(smoothed) $\mathbf{H}$ and $\mathbf{W}$, where the weights focus the orthogonality penalty preferentially on those factors that contribute the most power to the reconstruction. This concept of using power-weighted regularization penalties may be applicable more generally to matrix factorization techniques.

As in many data analysis scenarios, a variety of statistical approaches may be brought to bear on finding sequences in neural data. A classic method is to construct cross-correlogram plots, showing spike time correlations between pairs of neurons at various time lags. However, other forms of spike rate covariation, such as trial-to-trial gain modulation, can produce spurious peaks in this measure (*Brody, 1999*); recent work has developed statistical corrections for these effects (*Russo and Durstewitz, 2017*). After significant pairwise correlations are identified, one can heuristically piece together pairs of neurons with significant interactions into a sequence. This bottom-up approach may be better than seqNMF at detecting sequences involving small numbers of neurons, since seqNMF specifically targets sequences that explain large amounts of variance in the data. On the other hand, bottom-up approaches to sequence extraction may fail to identify long sequences with high participation noise or jitter in each neuron (*Quaglio et al., 2018*). One can think of seqNMF as a complementary top-down approach, which performs very well in the high-noise regime since it learns a template sequence at the level of the full population that is robust to noise at the level of individual units.

Statistical models with a dynamical component, such as Hidden Markov Models (HMMs) (*Maboudi et al., 2018*), linear dynamical systems (*Kao et al., 2015*), and models with switching dynamics (*Linderman et al., 2017*), can also capture sequential firing patterns. These methods will typically require many hidden states or latent dimensions to capture sequences, similar to PCA and NMF which require many components to recover sequences. Nevertheless, visualizing the transition matrix of an HMM can provide insight into the order in which hidden states of the model are visited, mapping onto different sequences that manifest in population activity (*Maboudi et al., 2018*). One advantage of this approach is that it can model sequences that occasionally end prematurely, while convNMF will always reconstruct the full sequence. On the other hand, this pattern completion property makes convNMF robust to participation noise and jitter. In contrast, a standard HMM must pass through each hidden state to model a sequence, and therefore may have trouble if many of these states are skipped. Thus, we expect HMMs and related models to exhibit complementary strengths and weaknesses when compared to convNMF.

Another strength of convNMF is its ability to accommodate sequences with shared neurons, as has been observed during song learning (*Okubo et al., 2015*). Sequences with shared neurons can be interpreted either in terms of 'parts-based' or 'events-based' factorizations (*Figure 9—figure supplement 2*). This capacity for a combinatorial description of overlapping sequences distinguishes convNMF from many other methods, which assume that neural patterns/sequences do not co-occur in time. For example, a vanilla HMM can only model each time step with a single hidden state and thus cannot express parts-based representations of neural sequences. Likewise, simple clustering models would assign each time interval to a single cluster label. Adding hierarchical and factorial structure to these models could allow them to test for overlapping neural sequences (see e.g. *Ghahramani and Jordan, 1997*); however, we believe seqNMF provides a simpler and more direct framework to explore this possibility.

Finally, as demonstrated by our development of new regularization terms and stability measures, convolutional matrix factorization is a flexible and extensible framework for sequence extraction. For example, one can tune the overall sparsity in the model by introducing additional L1 regularization terms. The loss function may also be modified, for example substituting in KL divergence or more general $\beta$-divergence (*Villasana et al., 2018*). Both L1 regularization and $\beta$-divergence losses are included in the seqNMF code package so that the model can be tuned to the particular needs of future analyses. Future development could incorporate outlier detection into the objective function (*Netrapalli et al., 2014*), or online optimization methods for large datasets (*Wang et al., 2013*). Other extensions to NMF, for example, Union of Intersections NMF Cluster (*Ubaru et al., 2017*), have yielded increased robustness and consistency of NMF factorizations, and could potentially also be modified for application to convNMF. Thus, adding convolutional structure to factorization-based models of neural data represents a rich opportunity for statistical neuroscience.

Despite limiting ourselves to a relatively simple model for the purposes of this paper, we extracted biological insights that would have been difficult to otherwise achieve. For example, we identified neural sequences in isolated songbirds without aligning to song syllables, enabling new

research into songbird learning on two fronts. First, since isolated and juvenile birds sing highly variable songs that are not easily segmented into stereotyped syllables, it is difficult and highly subjective to identify sequences by aligning to human-labeled syllables. SeqNMF enables the discovery and future characterization of neural sequences in these cases. Second, while behaviorally aligned sequences exist in tutored birds, it is that possible many neural sequences—for example, in different brain areas or stages of development—are not closely locked to song syllables. Thus, even in cases where stereotyped song syllables exist, behavioral alignment may overlook relevant sequences and structure in the data. These lessons apply broadly to many neural systems, and demonstrate the importance of general-purpose methods that extract sequences without reference to behavior. Our results show that convolutional matrix factorization models are an attractive framework to meet this need.

# Materials and methods

**Key resources table**

| Reagent type (species) or resource | Designation | Source or reference | Identifiers | Additional information |
|---|---|---|---|---|
| Software, algorithm | seqNMF | this paper | https://github.com/FeeLab/seqNMF | start with demo.m |
| Software, algorithm | convNMF | *Smaragdis, 2004*; *Smaragdis, 2007* | https://github.com/colinvaz/nmf-toolbox | |
| Software, algorithm | sparse convNMF | *O'Grady and Pearlmutter, 2006*; *Ramanarayanan et al., 2013* | https://github.com/colinvaz/nmf-toolbox | |
| Software, algorithm | NMF orthogonality penalties | *Choi, 2008*; *Chen and Cichocki, 2004* | | |
| Software, algorithm | other NMF extensions | *Cichocki et al., 2009* | | |
| Software, algorithm | NMF | *Lee and Seung, 1999* | | |
| Software, algorithm | CNMF_E (cell extraction) | *Zhou et al., 2018* | https://github.com/zhoupc/CNMF_E | |
| Software, algorithm | MATLAB | MathWorks | www.mathworks.com, RRID:SCR_001622 | |
| Strain, strain background (*adeno-associated virus*) | AAV9.CAG.GCaMP6f. WPRE.SV40 | *Chen et al., 2013* | Addgene viral prep # 100836-AAV9, http://n2t.net/addgene:100836, RRID:Addgene_100836 | |
| Commercial assay or kit | Miniature microscope | Inscopix | https://www.inscopix.com/nvista | |

## Contact for resource sharing

Further requests should be directed to Michale Fee (fee@mit.edu).

## Software and data availability

The seqNMF MATLAB code is publicly available as a github repository, which also includes our songbird data (*Figure 9*) for demonstration (*Mackevicius et al., 2018*; copy archived at https://github.com/elifesciences-publications/seqNMF).

The repository includes the seqNMF function, as well as helper functions for selecting $\lambda$, testing the significance of factors, plotting, and other functions. It also includes a demo script with an example of how to select $\lambda$ for a new dataset, test for significance of factors, plot the seqNMF factorization, switch between parts-based and events-based factorizations, and calculate cross-validated performance on a masked test set.

## Generating simulated data

We simulated neural sequences containing between 1 and 10 distinct neural sequences in the presence of various noise conditions. Each neural sequence was made up of 10 consecutively active

neurons, each separated by three timebins. The binary activity matrix was convolved with an exponential kernel ($\tau = 10$ timebins) to resemble neural calcium imaging activity.

## SeqNMF algorithm details

The x-ortho penalized convNMF algorithm is a direct extension of the multiplicative update convNMF algorithm (*Smaragdis, 2004*), and draws on previous work regularizing NMF to encourage factor orthogonality (*Chen and Cichocki, 2004*).

The uniqueness and consistency of traditional NMF has been better studied than convNMF. In special cases, NMF has a unique solution comprised of sparse, 'parts-based' features that can be consistently identified by known algorithms (*Donoho and Stodden, 2004*; *Arora et al., 2011*). However, this ideal scenario does not hold in many practical settings. In these cases, NMF is sensitive to initialization, resulting in potentially inconsistent features. This problem can be addressed by introducing additional constraints or regularization terms that encourage the model to extract particular, e.g. sparse or approximately orthogonal features (*Huang et al., 2014*; *Kim and Park, 2008*). Both theoretical work and empirical observations suggest that these modifications result in more consistently identified features (*Theis et al., 2005*; *Kim and Park, 2008*).

For x-ortho penalized seqNMF, we added to the convNMF cost function a term that promotes competition between overlapping factors, resulting in the following cost function:

$$(\widetilde{\mathbf{W}}, \widetilde{\mathbf{H}}) = \underset{\mathbf{W}, \mathbf{H}}{\arg\min} \left( ||\widetilde{\mathbf{X}} - \mathbf{X}||_F^2 + \lambda ||(\mathbf{W} \overset{\top}{\circledast} \mathbf{X})\mathbf{S}\mathbf{H}^\top||_{1, i \neq j} \right) \tag{8}$$

We derived the following multiplicative update rules for $\mathbf{W}$ and $\mathbf{H}$ (Appendix 1):

$$\mathbf{W}_{..\ell} \leftarrow \mathbf{W}_{..\ell} \times \frac{\mathbf{X}\left(\overset{\ell \rightarrow}{\mathbf{H}}\right)^\top}{\widetilde{\mathbf{X}}\left(\overset{\ell \rightarrow}{\mathbf{H}}\right)^\top + \lambda \overset{\leftarrow \ell}{\mathbf{X}} \mathbf{S}\mathbf{H}^\top (\mathbf{1} - \mathbf{I})} \tag{9}$$

$$\mathbf{H} \leftarrow \mathbf{H} \times \frac{\mathbf{W} \overset{\top}{\circledast} \mathbf{X}}{\mathbf{W} \overset{\top}{\circledast} \widetilde{\mathbf{X}} + \lambda (\mathbf{1} - \mathbf{I})(\mathbf{W} \overset{\top}{\circledast} \mathbf{X} \mathbf{S})} \tag{10}$$

where the division and $\times$ are element-wise. The operator $\overset{\ell \rightarrow}{(\cdot)}$ shifts a matrix in the $\rightarrow$ direction by $\ell$ timebins, that is a delay by $\ell$ timebins, and $\overset{\leftarrow \ell}{(\cdot)}$ shifts a matrix in the $\leftarrow$ direction by $\ell$ timebins (notation summary, *Table 1*). Note that multiplication with the $K \times K$ matrix $(\mathbf{1} - \mathbf{I})$ effectively implements factor competition because it places in the $k$th row a sum across all other factors. These update rules are derived in Appendix 1 by taking the derivative of the cost function in *Equation 8* and choosing an appropriate learning rate for each element.

In addition to the multiplicative updates outlined in *Table 2*, we also renormalize so rows of $\mathbf{H}$ have unit norm; shift factors to be centered in time such that the center of mass of each $\mathbf{W}$ pattern occurs in the middle; and in the final iteration run one additional step of unregularized convNMF to prioritize the cost of reconstruction error over the regularization (Algorithm 1). This final step is done to correct a minor suppression in the amplitude of some peaks in $\mathbf{H}$ that may occur within $2L$ timebins of neighboring sequences.

## Testing the significance of each factor on held-out data

In order to test whether a factor is significantly present in held-out data, we measured the distribution across timebins of the overlaps of the factor with the held-out data, and compared the skewness of this distribution to the null case (*Figure 1*). Overlap with the data is measured as $\mathbf{W} \overset{\top}{\circledast} \mathbf{X}$, a quantity which will be high at timepoints when the sequence occurs, producing a distribution of $\mathbf{W} \overset{\top}{\circledast} \mathbf{X}$ with high skew. In contrast, a distribution of overlaps exhibiting low skew indicates a sequence is not present in the data, since there are few timepoints of particularly high overlap. We estimated what skew levels would appear by chance by constructing null factors where temporal

relationships between neurons have been eliminated. To create such null factors, we start from the real factors then circularly shift the timecourse of each neuron by a random amount between 0 and $L$. We measure the skew of the overlap distributions for each null factor, and ask whether the skew we measured for the real factor is significant at p-value $\alpha$, that is, if it exceeds the Bonferroni corrected $((1 - \frac{\alpha}{K}) \times 100)^{th}$ percentile of the null skews (see *Figure 2—figure supplement 1*).

---

**Algorithm 1**: SeqNMF (x-ortho algorithm)

---

 **Input**: Data matrix $\mathbf{X}$, number of factors $K$, factor duration $L$, regularization
 strength $\lambda$
 **Output**: Factor exemplars $\mathbf{W}$, factor timecourses $\mathbf{H}$
1 Initialize $\mathbf{W}$ and $\mathbf{H}$ randomly
2 Iter = 1
3 **While** (Iter < maxIter) and ($\Delta$ cost > tolerance) **do**
4 Update $\mathbf{H}$ using multiplicative update from *Table 2*
5 Shift $\mathbf{W}$ and $\mathbf{H}$ to center $\mathbf{W}$'s in time
6 Renormalize $\mathbf{W}$ and $\mathbf{H}$ so rows of $\mathbf{H}$ have unit norm
7 Update $\mathbf{W}$ using multiplicative update from *Table 2*
8 Iter = Iter + 1
9 Do one final unregularized convNMF update of $\mathbf{W}$ and $\mathbf{H}$
10 **return**

---

Note that if $\lambda$ is set too small, seqNMF will produce multiple redundant factors to explain one sequence in the data. In this case, each redundant candidate sequence will pass the significance test outlined here. We will address below a procedure for choosing $\lambda$ and methods for determining the number of sequences.

## Calculating the percent power explained by a factorization

In assessing the relevance of sequences in a dataset, it can be useful to calculate what percentage of the total power in the dataset is explained by the factorization ($\widetilde{\mathbf{X}}$). The total power in the data is $\sum \mathbf{X}^2$ (abbreviating $\sum_{n=1}^{N} \sum_{t=1}^{T} x_{nt}^2$ to $\sum X^2$). The power unexplained by the factorization is $\sum (\mathbf{X} - \widetilde{\mathbf{X}})^2$. Thus, the percent of the total power explained by the factorization is:

$$\frac{\sum \mathbf{X}^2 - \sum (\mathbf{X} - \widetilde{\mathbf{X}})^2}{\sum \mathbf{X}^2} = \frac{\sum 2\mathbf{X}\widetilde{\mathbf{X}} - \widetilde{\mathbf{X}}^2}{\sum \mathbf{X}^2} \tag{11}$$

## 'Sequenciness' score

The 'sequenciness' score was developed to distinguish between datasets with exclusively synchronous patterns, and datasets with temporally extended sequential patterns. This score relies on the observation that synchronous patterns are not disrupted by shuffling the columns of the data matrix. The 'sequenciness' score is calculated by first computing the difference between the power explained by seqNMF in the actual and column-shuffled data. This quantity is then divided by the power explained in the actual data minus the power explained in data where each neuron is time-shuffled by a different random permutation.

## Choosing appropriate parameters for a new dataset

The choice of appropriate parameters ($\lambda$, $K$ and $L$) will depend on the data type (sequence length, number, and density; amount of noise; etc.).

In practice, we found that results were relatively robust to the choice of parameters. When $K$ or $L$ is set larger than necessary, seqNMF tends to simply leave the unnecessary factors or times empty. For choosing $\lambda$, the goal is to find the 'sweet spot' (*Figure 4*) to explain as much data as possible while still producing sensible factorizations, that is, minimally correlated factors, with low values of $||(\mathbf{W} \overset{\top}{\circledast} \mathbf{X}) \mathbf{S} \mathbf{H}^\top||_{1,i \neq j}$. Our software package includes demo code for determining the best parameters for a new type of data, using the following strategy:

1. Start with $K$ slightly larger than the number of sequences anticipated in the data
2. Start with $L$ slightly longer than the maximum expected factor length
3. Run seqNMF for a range of $\lambda$'s, and for each $\lambda$ measure the reconstruction error

$$\left( ||\mathbf{X} - \mathbf{W} \circledast \mathbf{H}||_F^2 \right) \text{ and the correlation cost term } \left( ||(\mathbf{W} \overset{\top}{\circledast} \mathbf{X})\mathbf{SH}^\top||_{1, i \neq j} \right)$$

4. Choose a $\lambda$ slightly above the crossover point $\lambda_0$
5. Decrease $K$ if desired, as otherwise some factors will be consistently empty
6. Decrease $L$ if desired, as otherwise some times will consistently be empty

In some applications, achieving the desired accuracy may depend on choosing a $\lambda$ that allows some inconsistency. It is possible to deal with this remaining inconsistency by comparing factors produced by different random initializations, and only considering factors that arise from several different initializations, a strategy that has been previously applied to standard convNMF on neural data (*Peter et al., 2017*).

During validation of our procedure for choosing $\lambda$, we compared factorizations to ground truth sequences as shown in *Figure 4*. To find the optimal value of $\lambda$, we used the product of two curves. The first curve was obtained by calculating the fraction of fits in which the true number of sequences was recovered as a function of $\lambda$. The second curve was obtained by calculating similarity to ground truth as a function of $\lambda$ (see Materials and methods section 'Measuring performance on noisy fits by comparing seqNMF sequence to ground-truth sequences'). The product of these two curves was smoothed using a three-sample boxcar sliding window, and the width was found as the values of $\lambda$ on either side of the peak value that correspond most closely to the half-maximum points of the curve.

## Preprocessing

While seqNMF is generally quite robust to noisy data, and different types of sequential patterns, proper preprocessing of the data can be important to obtaining reasonable factorizations on real neural data. A key principle is that, in minimizing the reconstruction error, seqNMF is most strongly influenced by parts of the data that exhibit high variance. This can be problematic if the regions of interest in the data have relatively low amplitude. For example, high firing rate neurons may be prioritized over those with lower firing rate. As an alternative to subtracting the mean firing rate of each neuron, which would introduce negative values, neurons could be normalized divisively or by subtracting off a NMF reconstruction fit using a method that forces a non-negative residual (*Kim and Smaragdis, 2014*). Additionally, variations in behavioral state may lead to seqNMF factorizations that prioritize regions of the data with high variance and neglect other regions. It may be possible to mitigate these effects by normalizing data, or by restricting analysis to particular subsets of the data, either by time or by neuron.

## Measuring performance on noisy data by comparing seqNMF sequences to ground-truth sequences

We wanted to measure the ability of seqNMF to recover ground-truth sequences even when the sequences are obstructed by noise. Our noisy data consisted of three ground-truth sequences, obstructed by a variety of noise types. For each ground-truth sequence, we found its best match among the seqNMF factors. This was performed in a greedy manner. Specifically, we first computed a reconstruction for one of the ground-truth factors. We then measured the correlation between this reconstruction and reconstructions generated from each of the extracted factors, and chose the best match (highest correlation). Next, we matched a second ground-truth sequence with its best match (highest correlation between reconstructions) among the remaining seqNMF factors, and finally we found the best match for the third ground-truth sequence. The mean of these three correlations was used as a measure of similarity between the seqNMF factorization and the ground-truth (noiseless) sequences.

## Testing generalization of factorization to randomly held-out (masked) data entries

The data matrix $\mathbf{X}$ was divided into training data and test data by randomly selecting 5 or 10% of matrix entries to hold out. Specifically, the objective function (*Equation 5*, in the Results section) was modified to:

$$\underset{\mathbf{W},\mathbf{H}}{\arg\min}\,||\mathbf{M}\times(\mathbf{W}\circledast\mathbf{H}-\mathbf{X})||_F^2+\mathscr{R} \tag{12}$$

where $\times$ indicates elementwise multiplication (Hadamard product) and $\mathbf{M}$ is a binary matrix with 5 or 10% of the entries randomly selected to be zero (held-out test set) and the remaining 95 or 90% set to one (training set). To search for a solution, we reformulate this optimization problem as:

$$\underset{\mathbf{W},\mathbf{H},\mathbf{Z}}{\arg\min}\,||\mathbf{W}\circledast\mathbf{H}-\mathbf{Z}||_F^2+\mathscr{R}$$
$$\text{subject to}\,\mathbf{M}\times\mathbf{Z}=\mathbf{M}\times\mathbf{X} \tag{13}$$

where we have introduced a new optimization variable $\mathbf{Z}$, which can be thought of as a surrogate dataset that is equal to the ground truth data only on the training set. The goal is now to minimize the difference between the model estimate, $\widetilde{\mathbf{X}}=\mathbf{W}\circledast\mathbf{H}$, and the surrogate, $\mathbf{Z}$, while constraining $\mathbf{Z}$ to equal $\mathbf{X}$ at unmasked elements (where $m_{ij}=1$) and allowing $\mathbf{Z}$ to be freely chosen at masked elements (where $m_{ij}=0$). Clearly, at masked elements, the best choice is to make $\mathbf{Z}$ equal to the current model estimate $\widetilde{\mathbf{X}}$ as this minimizes the cost function without violating the constraint. This leads to the following update rules which are applied cyclically to update $\mathbf{Z}$, $\mathbf{W}$, and $\mathbf{H}$.

$$\mathbf{Z}_{nt}\leftarrow\begin{cases}\mathbf{X}_{nt} & \text{if}\quad \mathbf{M}_{nt}=1\\(\mathbf{W}\circledast\mathbf{H})_{nt} & \text{if}\quad \mathbf{M}_{nt}=0\end{cases} \tag{14}$$

$$\mathbf{W}_{\cdot\cdot\ell}\leftarrow\mathbf{W}_{\cdot\cdot\ell}\times\frac{\mathbf{Z}\left(\overset{\ell\rightarrow}{\mathbf{H}}\right)^\top}{\widetilde{\mathbf{X}}\left(\overset{\ell\rightarrow}{\mathbf{H}}\right)^\top+\lambda\,\overset{\leftarrow\ell}{\mathbf{Z}}\mathbf{S}\mathbf{H}^\top(\mathbf{1}-\mathbf{I})} \tag{15}$$

$$\mathbf{H}\leftarrow\mathbf{H}\times\frac{\overset{\top}{\mathbf{W}}\circledast\mathbf{Z}}{\overset{\top}{\mathbf{W}}\circledast\widetilde{\mathbf{X}}+\lambda(\mathbf{1}-\mathbf{I})(\overset{\top}{\mathbf{W}}\circledast\mathbf{Z}\mathbf{S})} \tag{16}$$

The measure used for testing generalization performance was root mean squared error (RMSE). For the testing phase, RMSE was computed from the difference between $\widetilde{\mathbf{X}}$ and the data matrix $\mathbf{X}$ only for held-out entries.

## Hippocampus data

The hippocampal data was collected in the Buzsaki lab (*Pastalkova et al., 2015*; *Mizuseki et al., 2013*), and is publicly available on the Collaborative Research in Computational Neuroscience (CRCNS) Data sharing website. The dataset we refer to as 'Rat 1' is in the hc-5 dataset, and the dataset we refer to as 'Rat 2' is in the hc-3 dataset. Before running seqNMF, we processed the data by convolving the raw spike trains with a gaussian kernel of standard deviation 100 ms.

## Animal care and use

We used male zebra finches (*Taeniopygia guttata*) from the MIT zebra finch breeding facility (Cambridge, MA). Animal care and experiments were carried out in accordance with NIH guidelines, and reviewed and approved by the Massachusetts Institute of Technology Committee on Animal Care (protocol 0715-071-18).

In order to prevent exposure to a tutor song, birds were foster-raised by female birds, which do not sing, starting on or before post-hatch day 15. For experiments, birds were housed singly in custom-made sound isolation chambers.

## Data acquisition and preprocessing

The calcium indicator GCaMP6f was expressed in HVC by intracranial injection of the viral vector AAV9.CAG.GCaMP6f.WPRE.SV40 (*Chen et al., 2013*) into HVC. In the same surgery, a cranial window was made using a GRIN (gradient index) lens (1 mm diameter, 4 mm length, Inscopix). After at

least one week, in order to allow for sufficient viral expression, recordings were made using the Inscopix nVista miniature fluorescent microscope.

Neuronal activity traces were extracted from raw fluorescence movies using the CNMF_E algorithm, a constrained non-negative matrix factorization algorithm specialized for microendoscope data by including a local background model to remove activity from out-of-focus cells (*Zhou et al., 2018*).

We performed several preprocessing steps before applying seqNMF to functional calcium traces extracted by CNMF_E. First, we estimated burst times from the raw traces by deconvolving the traces using an AR-2 process. The deconvolution parameters (time constants and noise floor) were estimated for each neuron using the CNMF_E code package (*Zhou et al., 2018*). Some neurons exhibited larger peaks than others, likely due to different expression levels of the calcium indicator. Since seqNMF would prioritize the neurons with the most power, we renormalized by dividing the signal from each neuron by the sum of the maximum value of that row and the $95^{th}$ percentile of the signal across all neurons. In this way, neurons with larger peaks were given some priority, but not much more than that of neurons with weaker signals.

## Acknowledgements

This work was supported by a grant from the Simons Collaboration for the Global Brain, the National Institutes of Health (NIH) [grant number R01 DC009183] and the G Harold and Leila Y Mathers Charitable Foundation. ELM received support through the NDSEG Fellowship program. AHB received support through NIH training grant 5T32EB019940-03. MSG received support from the NIH [grant number U19NS104648]. AHW received support from the U.S. Department of Energy Computational Science Graduate Fellowship (CSGF) program. Thanks to Pengcheng Zhou for advice on his CNMF_E calcium data cell extraction algorithm. Thanks to Wiktor Młynarski for helpful convNMF discussions. Thanks to Michael Stetner, Galen Lynch, Nhat Le, Dezhe Jin, Edward Nieh, Adam Charles, Jane Van Velden and Yiheng Wang for comments on the manuscript and on our code package. Thanks to our reviewers for wonderful suggestions, including the use of *diss* to select $K$, and using seqNMF to measure 'sequenciness'. Special thanks to the 2017 Methods in Computational Neuroscience course [supported by NIH grant R25 MH062204 and Simons Foundation] at the Woods Hole Marine Biology Lab, where this collaboration was started.

## Additional information

### Funding

| Funder | Grant reference number | Author |
|---|---|---|
| Simons Foundation | Simons Collaboration for the Global Brain | Mark S Goldman Michale S Fee |
| National Institute on Deafness and Other Communication Disorders | R01 DC009183 | Michale S Fee |
| G Harold and Leila Y. Mathers Foundation | | Michale S Fee |
| U.S. Department of Defense | NDSEG Fellowship program | Emily L Mackevicius |
| Department of Energy, Labor and Economic Growth | Computational Science Graduate Fellowship (CSGF) | Alex H Williams |
| NIH Office of the Director | Training Grant 5T32EB019940-03 | Andrew H Bahle |
| National Institute of Neurological Disorders and Stroke | U19 NS104648 | Mark S Goldman |
| National Institute of Mental Health | R25 MH062204 | Mark S Goldman Michale S Fee |

The funders had no role in study design, data collection and interpretation, or the decision to submit the work for publication.

## Author contributions
Emily L Mackevicius, Andrew H Bahle, Conceptualization, Data curation, Software, Formal analysis, Validation, Investigation, Visualization, Methodology, Writing—original draft, Writing—review and editing; Alex H Williams, Conceptualization, Formal analysis, Validation, Methodology, Writing—review and editing; Shijie Gu, Data curation; Natalia I Denisenko, Investigation; Mark S Goldman, Conceptualization, Formal analysis, Supervision, Funding acquisition, Methodology, Writing—original draft, Project administration, Writing—review and editing; Michale S Fee, Conceptualization, Resources, Formal analysis, Supervision, Funding acquisition, Validation, Visualization, Methodology, Writing—original draft, Project administration, Writing—review and editing

## Author ORCIDs
Emily L Mackevicius http://orcid.org/0000-0001-6593-4398
Andrew H Bahle http://orcid.org/0000-0003-0567-7195
Alex H Williams https://orcid.org/0000-0001-5853-103X
Shijie Gu http://orcid.org/0000-0001-6257-5756
Mark S Goldman http://orcid.org/0000-0002-8257-2314
Michale S Fee http://orcid.org/0000-0001-7539-1745

## Ethics
Animal experimentation: Animal care and experiments were carried out in accordance with NIH guidelines, and reviewed and approved by the Massachusetts Institute of Technology Committee on Animal Care (protocol 0715-071-18).

## Decision letter and Author response
Decision letter https://doi.org/10.7554/eLife.38471.040
Author response https://doi.org/10.7554/eLife.38471.041

# Additional files

## Supplementary files
• Transparent reporting form
DOI: https://doi.org/10.7554/eLife.38471.031

## Data availability
Code and songbird dataset is publicly available on github: https://github.com/FeeLab/seqNMF (copy archived at https://github.com/elifesciences-publications/seqNMF). Rat datasets were collected in the Buzsaki lab, and are publicly available on CRCNS (https://crcns.org/data-sets/hc); users must first create a free account (https://crcns.org/register) before they can download the datasets from the site.

The following previously published datasets were used:

| Author(s) | Year | Dataset title | Dataset URL | Database and Identifier |
|---|---|---|---|---|
| Mizuseki K, Sirota A, Pastalkova E, Diba K | 2013 | Multiple single unit recordings from different rat hippocampal and entorhinal regions while the animals were performing multiple behavioral tasks. CRCNS.org. | https://crcns.org/data-sets/hc/hc-3 | Collaborative Research in Computational Neuroscience, 10.60 80/K09G5JRZ |
| Pastalkova E, Wang Y | 2015 | Simultaneous extracellular recordings from left and right hippocampal areas CA1 and right entorhinal cortex from a rat performing a left / right alternation task and other behaviors. CRCNS.org. | https://crcns.org/data-sets/hc/hc-5 | Collaborative Research in Computational Neuroscience, 10.60 80/K0KS6PHF |

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

## Appendix 1

DOI: https://doi.org/10.7554/eLife.38471.032

### Deriving multiplicative update rules

Standard gradient descent methods for minimizing a cost function must be adapted when solutions are constrained to be non-negative, since gradient descent steps may result in negative values. Lee and Seung invented an elegant and widely-used algorithm for non-negative gradient descent that avoids negative values by performing multiplicative updates (*Lee and Seung, 2001*; *Lee and Seung, 1999*). They derived these multiplicative updates by choosing an adaptive learning rate that makes additive terms cancel from standard gradient descent on the cost function. We will reproduce their derivation here, and detail how to extend it to the convolutional case (*Smaragdis, 2004*) and apply several forms of regularization (*O'Grady and Pearlmutter, 2006*; *Ramanarayanan et al., 2013*; *Chen and Cichocki, 2004*). See *Table 2* for a compilation of cost functions, derivatives and multiplicative updates for NMF and convNMF under several different regularization conditions.

#### Standard NMF

NMF performs the factorization $\mathbf{X} \approx \widetilde{\mathbf{X}} = \mathbf{WH}$. NMF factorizations seek to solve the following problem:

$$(\widetilde{\mathbf{W}}, \widetilde{\mathbf{H}}) \;=\; arg\min_{\mathbf{W},\mathbf{H}} \mathscr{L}(\mathbf{W}, \mathbf{H}) \tag{17}$$

$$\mathscr{L}(\mathbf{W}, \mathbf{H}) \;=\; \frac{1}{2}\|\widetilde{\mathbf{X}} - \mathbf{X}\|_F^2 \tag{18}$$

$$\widetilde{\mathbf{W}}, \widetilde{\mathbf{H}} \geq 0 \tag{19}$$

This problem is convex in $\mathbf{W}$ and $\mathbf{H}$ separately, not together, so a local minimum is found by alternating $\mathbf{W}$ and $\mathbf{H}$ updates. Note that:

$$\frac{\partial}{\partial \mathbf{W}} \mathscr{L}(\mathbf{W}, \mathbf{H}) \;=\; \widetilde{\mathbf{X}}\mathbf{H}^\top - \mathbf{X}\mathbf{H}^\top \tag{20}$$

$$\frac{\partial}{\partial \mathbf{H}} \mathscr{L}(\mathbf{W}, \mathbf{H}) \;=\; \mathbf{W}^\top\widetilde{\mathbf{X}} - \mathbf{W}^\top\mathbf{X} \tag{21}$$

Thus, gradient descent steps for $\mathbf{W}$ and $\mathbf{H}$ are:

$$\mathbf{W} \leftarrow \mathbf{W} - \eta_{\mathbf{W}}(\widetilde{\mathbf{X}}\mathbf{H}^\top - \mathbf{X}\mathbf{H}^\top) \tag{22}$$

$$\mathbf{H} \leftarrow \mathbf{H} - \eta_{\mathbf{H}}(\mathbf{W}^\top\widetilde{\mathbf{X}} - \mathbf{W}^\top\mathbf{X}) \tag{23}$$

To arrive at multiplicative updates, Lee and Seung (*Lee and Seung, 2001*) set:

$$\eta_{\mathbf{W}} \;=\; \frac{\mathbf{W}}{\mathbf{WHH}^\top} \tag{24}$$

$$\eta_{\mathbf{H}} \;=\; \frac{\mathbf{H}}{\mathbf{W}^\top\mathbf{WH}} \tag{25}$$

Thus, the gradient descent updates become multiplicative:

$$\mathbf{W} \leftarrow \mathbf{W} \times \frac{\mathbf{XH}^\top}{\mathbf{WHH}^\top} = \mathbf{W} \times \frac{\mathbf{XH}^\top}{\widetilde{\mathbf{X}}\mathbf{H}^\top} \tag{26}$$

$$\mathbf{H} \leftarrow \mathbf{H} \times \frac{\mathbf{W}^\top \mathbf{X}}{\mathbf{W}^\top \mathbf{WH}} = \mathbf{H} \times \frac{\mathbf{W}^\top \mathbf{X}}{\mathbf{W}^\top \widetilde{\mathbf{X}}} \tag{27}$$

where the division and $\times$ are element-wise.

## Standard convNMF

Convolutional NMF factorizes data $\mathbf{X} \approx \widetilde{\mathbf{X}} = \sum_\ell \mathbf{W}_{..\ell} \overset{\ell\rightarrow}{\mathbf{H}} = \mathbf{W} \circledast \mathbf{H}$. ConvNMF factorizations seek to solve the following problem:

$$(\widetilde{\mathbf{W}}, \widetilde{\mathbf{H}}) = arg\min_{\mathbf{W},\mathbf{H}} \mathscr{L}(\mathbf{W},\mathbf{H}) \tag{28}$$

$$\mathscr{L}(\mathbf{W},\mathbf{H}) = \frac{1}{2}||\widetilde{\mathbf{X}} - \mathbf{X}||_F^2 \tag{29}$$

$$\widetilde{\mathbf{W}}, \widetilde{\mathbf{H}} \geq 0 \tag{30}$$

The derivation above for standard NMF can be applied for each $\ell$, yielding the following update rules for convNMF (**Smaragdis, 2004**):

$$\mathbf{W}_{..\ell} \leftarrow \mathbf{W}_{..\ell} \times \frac{\mathbf{X}\overset{\ell\rightarrow}{\mathbf{H}}^\top}{\widetilde{\mathbf{X}}\overset{\ell\rightarrow}{\mathbf{H}}^\top} \tag{31}$$

$$\mathbf{H} \leftarrow \mathbf{H} \times \frac{\sum_\ell \mathbf{W}_{..\ell}^\top \overset{\leftarrow\ell}{\mathbf{X}}}{\sum_\ell \mathbf{W}_{..\ell}^\top \overset{\leftarrow\ell}{\widetilde{\mathbf{X}}}} = \mathbf{H} \times \frac{\mathbf{W} \overset{\top}{\circledast} \mathbf{X}}{\mathbf{W} \overset{\top}{\circledast} \widetilde{\mathbf{X}}} \tag{32}$$

Where the operator $\ell \rightarrow$ shifts a matrix in the $\rightarrow$ direction by $\ell$ timebins, that is a delay by $\ell$ timebins, and $\leftarrow \ell$ shifts a matrix in the $\leftarrow$ direction by $\ell$ timebins (**Table 1**). Note that NMF is a special case of convNMF where $L = 1$.

## Incorporating regularization terms

Suppose we want to regularize by adding a new term, $\mathscr{R}$ to the cost function:

$$(\widetilde{\mathbf{W}}, \widetilde{\mathbf{H}}) = arg\min_{\mathbf{W},\mathbf{H}} \mathscr{L}(\mathbf{W},\mathbf{H}) \tag{33}$$

$$\mathscr{L}(\mathbf{W},\mathbf{H}) = \frac{1}{2}||\widetilde{\mathbf{X}} - \mathbf{X}||_F^2 + \mathscr{R} \tag{34}$$

$$\widetilde{\mathbf{W}}, \widetilde{\mathbf{H}} \geq 0 \tag{35}$$

Using a similar trick to Lee and Seung, we choose a $\eta_\mathbf{W}, \eta_\mathbf{H}$ to arrive at a simple multiplicative update. Below is the standard NMF case, which generalizes trivially to the convNMF case.

Note that:

$$\frac{\partial \mathcal{L}}{\partial \mathbf{W}} = \widetilde{\mathbf{X}}\mathbf{H}^\top - \mathbf{X}\mathbf{H}^\top + \frac{\partial \mathcal{R}}{\partial \mathbf{W}} \qquad (36)$$

$$\frac{\partial \mathcal{L}}{\partial \mathbf{H}} = \mathbf{W}^\top \widetilde{\mathbf{X}} - \mathbf{W}^\top \mathbf{X} + \frac{\partial \mathcal{R}}{\partial \mathbf{H}} \qquad (37)$$

We set:

$$\eta_{\mathbf{W}} = \frac{\mathbf{W}}{\widetilde{\mathbf{X}}\mathbf{H}^\top + \frac{\partial \mathcal{R}}{\partial \mathbf{W}}} \qquad (38)$$

$$\eta_{\mathbf{H}} = \frac{\mathbf{H}}{\mathbf{W}^\top \widetilde{\mathbf{X}} + \frac{\partial \mathcal{R}}{\partial \mathbf{H}}} \qquad (39)$$

Thus, the gradient descent updates become multiplicative:

$$\mathbf{W} \leftarrow \mathbf{W} - \eta_{\mathbf{W}} \frac{\partial \mathcal{L}}{\partial \mathbf{W}} = \mathbf{W} \times \frac{\mathbf{X}\mathbf{H}^\top}{\widetilde{\mathbf{X}}\mathbf{H}^\top + \frac{\partial \mathcal{R}}{\partial \mathbf{W}}} \qquad (40)$$

$$\mathbf{H} \leftarrow \mathbf{H} - \eta_{\mathbf{H}} \frac{\partial \mathcal{L}}{\partial \mathbf{H}} = \mathbf{H} \times \frac{\mathbf{W}^\top \mathbf{X}}{\mathbf{W}^\top \widetilde{\mathbf{X}} + \frac{\partial \mathcal{R}}{\partial \mathbf{H}}} \qquad (41)$$

where the division and $\times$ are element-wise.

The above formulation enables flexible incorporation of different types of regularization or penalty terms into the multiplicative NMF update algorithm. This framework also extends naturally to the convolutional case. See **Table 2** for examples of several regularization terms, including $L1$ sparsity (**O'Grady and Pearlmutter, 2006**; **Ramanarayanan et al., 2013**) and spatial decorrelation (**Chen and Cichocki, 2004**), as well as the terms we introduce here to combat the types of inefficiencies and cross correlations we identified in convolutional NMF, namely, smoothed orthogonality for $\mathbf{H}$ and $\mathbf{W}$, and the x-ortho penalty term. For the x-ortho penalty term, $\lambda||(\mathbf{W}\overset{\top}{\circledast}\mathbf{X})\mathbf{S}\mathbf{H}^\top||_{1,i\neq j}$, the multiplicative update rules are:

$$\mathbf{W}_{\cdot\cdot\ell} \leftarrow \mathbf{W}_{\cdot\cdot\ell} \times \frac{\mathbf{X}\left(\overset{\ell\rightarrow}{\mathbf{H}}\right)^\top}{\widetilde{\mathbf{X}}\left(\overset{\ell\rightarrow}{\mathbf{H}}\right)^\top + \lambda \overset{\leftarrow\ell}{\mathbf{X}}\mathbf{S}\mathbf{H}^\top(\mathbf{1} - \mathbf{I})} \qquad (42)$$

$$\mathbf{H} \leftarrow \mathbf{H} \times \frac{\mathbf{W}\overset{\top}{\circledast}\mathbf{X}}{\mathbf{W}\overset{\top}{\circledast}\widetilde{\mathbf{X}} + \lambda(\mathbf{1} - \mathbf{I})(\mathbf{W}\overset{\top}{\circledast}\mathbf{X}\mathbf{S})} \qquad (43)$$

where the division and $\times$ are element-wise. Note that multiplication with the $K \times K$ matrix $(\mathbf{1} - \mathbf{I})$ effectively implements factor competition because it places in the $k$th row a sum across all other factors.

## Appendix 2

DOI: https://doi.org/10.7554/eLife.38471.032

# Relation of the x-ortho penalty to traditional regularizations

As noted in the main text, the x-ortho penalty term is not formally a regularization because it includes the data $\mathbf{X}$. In this Appendix, we show how this penalty can be approximated by a data-free regularization. The resulting regularization contains three terms corresponding to a weighted orthogonality penalty on pairs of $\mathbf{H}$ factors, a weighted orthogonality penalty on pairs of $\mathbf{W}$ factors, and a term that penalizes interactions among triplets of factors. We analyze each term in both the time domain (*Equation 50*) and in the frequency domain (*Equations 50 and 69*).

## Time domain analysis

We consider the cross-orthogonality penalty term:

$$\mathcal{R} = \|(\mathbf{W} \overset{\top}{\circledast} \mathbf{X})\mathbf{S}\mathbf{H}^\top\|_{1, i \neq j} \tag{44}$$

and define, $\mathbf{R} = (\mathbf{W} \overset{\top}{\circledast} \mathbf{X})\mathbf{S}\mathbf{H}^\top$, which is a $K \times K$ matrix. Each element $\mathbf{R}_{ij}$ is a positive number describing the overlap or correlation between factor $i$ and factor $j$ in the model. Each element of $\mathbf{R}$ can be written explicitly as:

$$\mathbf{R}_{ij} = \sum_t \sum_n \sum_\ell \mathbf{W}_{ni\ell} \mathbf{X}_{n(t+\ell)} \sum_\tau \mathbf{S}_{t\tau} \mathbf{H}_{j\tau} \tag{45}$$

Where the index variables $t$ and $\tau$ range from 1 to $T$, $n$ ranges from 1 to $N$, and $\ell$ ranges from 1 to $L$.

Our goal here is to find a close approximation to this penalty term that does not contain the data $\mathbf{X}$. This can readily be done if $\mathbf{X}$ is well-approximated by the convNMF decomposition:

$$\mathbf{X}_{nt} \approx (\mathbf{W} \circledast \mathbf{H})_{nt} = \sum_k \sum_\ell \mathbf{W}_{nk\ell} \mathbf{H}_{k(t-\ell)} \tag{46}$$

Substituting this expression into *Equation 45* and defining the smoothed matrix $\sum_\tau \mathbf{S}_{t\tau}\mathbf{H}_{j\tau}$ as $\mathbf{H}_{jt}^{\mathrm{smooth}}$ gives:

$$\mathbf{R}_{ij} \approx \sum_t \sum_n \sum_\ell \sum_k \sum_{\ell'} \mathbf{W}_{ni\ell} \mathbf{W}_{nk\ell'} \mathbf{H}_{kt} \mathbf{H}_{jt}^{\mathrm{smooth}} \tag{47}$$

Making the substitution $u = \ell - \ell'$ gives:

$$\mathbf{R}_{ij} \approx \sum_t \sum_n \sum_{u=-(L-1)}^{L-1} \sum_k \sum_{\ell'} \mathbf{W}_{ni(\ell'+u)} \mathbf{W}_{nk\ell'} \mathbf{H}_{k(t+u)} \mathbf{H}_{jt}^{\mathrm{smooth}} \tag{48}$$

where in the above expression we have taken $u = \ell - \ell'$ to extend over the full range from $-(L-1)$ to $(L-1)$ under the implicit assumption that $\mathbf{W}$ and $\mathbf{H}$ are zero padded such that values of $\mathbf{W}$ for lag indices outside the range 0 to $L-1$ and values of $\mathbf{H}$ for time indices outside the range 1 to $T$ are taken to be zero.

Relabeling $\ell'$ as $\ell$ and gathering terms together yields

$$\mathbf{R}_{ij} \approx \sum_k \sum_{u=-(L-1)}^{L-1} \left( \sum_n \sum_\ell \mathbf{W}_{ni(\ell+u)} \mathbf{W}_{nk\ell} \right) \left( \sum_t \mathbf{H}_{k(t+u)} \mathbf{H}_{jt}^{\text{smooth}} \right) \tag{49}$$

We note that the above expression contains terms that resemble penalties on orthogonality between two $\mathbf{W}$ factors (first parenthetical) or two $\mathbf{H}$ factors (one of which is smoothed, second parenthetical), but in this case allowing for different time lags $u$ between the factors. To understand this formula better, we decompose the above sum over $k$ into three contributions corresponding to $k = i$, $k = j$, and $k \neq i, j$

$$\begin{aligned}
\mathbf{R}_{ij} \approx &\sum_{u=-(L-1)}^{L-1} \left( \sum_n \sum_\ell \mathbf{W}_{ni(\ell+u)} \mathbf{W}_{ni\ell} \right) \left( \sum_t \mathbf{H}_{i(t+u)} \mathbf{H}_{jt}^{\text{smooth}} \right) + \\
&\sum_{u=-(L-1)}^{L-1} \left( \sum_n \sum_\ell \mathbf{W}_{ni(\ell+u)} \mathbf{W}_{nj\ell} \right) \left( \sum_t \mathbf{H}_{j(t+u)} \mathbf{H}_{jt}^{\text{smooth}} \right) + \\
&\sum_{k \neq i,j} \sum_{u=-(L-1)}^{L-1} \left( \sum_n \sum_\ell \mathbf{W}_{ni(\ell+u)} \mathbf{W}_{nk\ell} \right) \left( \sum_t \mathbf{H}_{k(t+u)} \mathbf{H}_{jt}^{\text{smooth}} \right)
\end{aligned} \tag{50}$$

The first term above contains, for $u = 0$, a simple extension of the $(i,j)^{th}$ element of the $\mathbf{H}$ orthogonality condition $\mathbf{HSH}^\top$. The extension is that the orthogonality is weighted by the power, that is the sum of squared elements, in the $i^{th}$ factor of $\mathbf{W}$ (the apparent lack of symmetry in weighing by the $i^{th}$ rather than the $j^{th}$ factor can be removed by simultaneously considering the term $R_{ji}$, as shown in the Fourier representation of the following section). This weighting has the benefit of applying the penalty on $\mathbf{H}$ orthogonality most strongly to those factors whose corresponding $\mathbf{W}$ components contain the most power. For $u \neq 0$, this orthogonality condition is extended to allow for overlap of time-shifted $\mathbf{H}$'s, with weighting at each time shift by the autocorrelation of the corresponding $\mathbf{W}$ factor. Qualitatively, this enforces that (even in the absence of the smoothing matrix $\mathbf{S}$), $\mathbf{H}$'s that are offset by less than the width of the autocorrelation of the corresponding $\mathbf{W}$'s will have overlapping convolutions with these $\mathbf{W}$'s due to the temporal smoothing associated with the convolution operation. We note that, for sparse sequences as in the examples of *Figure 1*, there is no time-lagged component to the autocorrelation, so this term corresponds simply to a smoothed $\mathbf{H}$ orthogonality regularization, weighted by the strength of the corresponding $\mathbf{W}$ factors.

The second term above represents a complementary orthogonality condition on the $\mathbf{W}$ components, in which orthogonality in the $i^{th}$ and $j^{th}$ factors are weighted by the (smoothed) autocorrelation of the $\mathbf{H}$ factors. For the case in which the $\mathbf{H}$ factors have no time-lagged autocorrelations, this corresponds to a simple weighting of $\mathbf{W}$ orthogonality by the strength of the corresponding $\mathbf{H}$ factors.

Finally, we consider the remaining terms of the cost function, for which $k \neq i, j$. We note that these terms are only relevant when the factorization contains at least three factors, and thus their role cannot be visualized from the simple Type 1 to Type 3 examples of *Figure 1*. These terms have the form:

$$\sum_{k \neq i,j} \sum_{u=-(L-1)}^{L-1} \left( \sum_n \sum_\ell \mathbf{W}_{ni(\ell+u)} \mathbf{W}_{nk\ell} \right) \left( \sum_t \mathbf{H}_{k(t+u)} \mathbf{H}_{jt}^{\text{smooth}} \right) \tag{51}$$

To understand how this term contributes, we consider each of the expressions in parentheses. The first expression corresponds, as described above, to the time-lagged cross correlation of the $i^{th}$ and $k^{th}$ $\mathbf{W}$ components. Likewise, the second expression corresponds to the time-lagged correlation of the (smoothed) $j^{th}$ and $k^{th}$ $\mathbf{H}$ components. Thus, this term of $\mathbf{R}_{ij}$ contributes whenever there is a factor $(\mathbf{W}_{\cdot k \cdot}, \mathbf{H}_{k \cdot})$ that overlaps, at the same time lags, with the $i^{th}$ factor's $\mathbf{W}$ component and the $j^{th}$ factor's $\mathbf{H}$ component. Thus, this term penalizes cases where, rather than (or in addition to) two factors $i$ and $j$ directly overlapping one another, they have a common factor $k$ with which they overlap.

An example of the contribution of a triplet penalty term, as well as of the paired terms of *Equation 50*, is shown in *Figure 1* of this Appendix. By inspection, there is a penalty $\mathbf{R}_{23}$ due to the overlapping values of the pair $(\mathbf{h}_2, \mathbf{h}_3)$. Likewise, there is a penalty $\mathbf{R}_{13}$ due to the overlapping values of the pair $(\mathbf{w}_1, \mathbf{w}_3)$. The triplet penalty term contributes to $\mathbf{R}_{12}$ and derives from the fact that $\mathbf{w}_1$ overlaps with $\mathbf{w}_3$ at the same time (and with the same, zero time lag) as $\mathbf{h}_2$ overlaps with $\mathbf{h}_3$.

In summary, the above analysis shows that for good reconstructions of the data where $\mathbf{X} \approx \mathbf{W} \circledast \mathbf{H}$, the x-ortho penalty can be well-approximated by the sum of three contributions. The first corresponds to a penalty on time-lagged (smoothed) $\mathbf{H}$ orthogonality weighted at each time lag by the autocorrelation of the corresponding $\mathbf{W}$ factors. The second similarly corresponds to a penalty on time-lagged $\mathbf{W}$ orthogonality weighted at each time lag by the (smoothed) autocorrelation of the corresponding $\mathbf{H}$ factors. For simple cases of sparse sequences, these contributions reduce to orthogonality in $\mathbf{H}$ or $\mathbf{W}$ weighted by the power in the corresponding $\mathbf{W}$ or $\mathbf{H}$, respectively, thus focusing the penalties most heavily on those factors which contribute most heavily to the data reconstruction. The third, triplet contribution corresponds to the case in which a factor in $\mathbf{W}$ and a different factor in $\mathbf{H}$ both overlap (at the same time lag) with a third common factor, and may occur even when the factors $\mathbf{W}$ and $\mathbf{H}$ themselves are orthogonal. Further work is needed to determine whether this third contribution is critical to the x-ortho penalty or is simply a by-product of the x-ortho penalty procedure's direct use of the data $\mathbf{X}$.

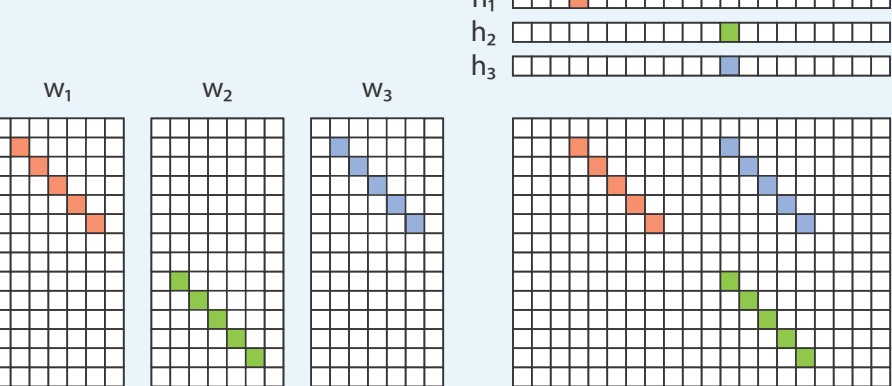

**Appendix 2—figure 1.** Example of redundancy with three factors. In addition to the direct overlap of $\mathbf{h}_2$ and $\mathbf{h}_3$, and of $\mathbf{w}_1$ and $\mathbf{w}_3$, there is a 'triplet' penalty $\mathbf{R}_{12}$ on factors 1 and 2 that occurs because each has an overlap (in either $\mathbf{W}$ or $\mathbf{H}$) with the 3rd factor $(\mathbf{w}_3, \mathbf{h}_3)$. This occurs even though neither $\mathbf{w}_1$ and $\mathbf{w}_2$, nor $\mathbf{h}_1$ and $\mathbf{h}_2$, are themselves overlapping.
DOI: https://doi.org/10.7554/eLife.38471.034

## Frequency domain analysis

Additional insight may be obtained by analyzing these three components of $\mathbf{R}$ in the Fourier domain. Before doing so, we below derive the Fourier domain representation of $\mathbf{R}$, and provide insights suggested by this perspective.

### Fourier representation of the x-ortho penalty

As in the time domain analysis, we start with defining:

$$\mathbf{R}_{ij} = \sum_t \sum_n \sum_\ell \mathbf{W}_{ni\ell} \mathbf{X}_{n(t+\ell)} \sum_\tau \mathbf{S}_{t\tau} \mathbf{H}_{j\tau} \tag{52}$$

Unpacking the notation above, we note that:

$$\mathbf{R}_{ij} = \left[ \sum_{n=1}^{N} Corr(\overrightarrow{\mathbf{W}}_i^{(n)}, \overrightarrow{\mathbf{X}}^{(n)}) \right] \cdot Conv(\overrightarrow{\mathbf{H}}_j, \overrightarrow{\mathbf{s}}) \tag{53}$$

where $\overrightarrow{\mathbf{W}}_i^{(n)}$ is the $n^{th}$ row of $\mathbf{W}_{\cdot i \cdot}$, $\overrightarrow{\mathbf{X}}^{(n)}$ is the $n^{th}$ row of $\mathbf{X}$, $\overrightarrow{\mathbf{H}}_j$ is $\mathbf{H}_{j \cdot}$, $\overrightarrow{\mathbf{s}}$ is a smoothing vector corresponding to the entries of each row of the smoothing matrix $\mathbf{S}$, and "·" is a dot product. For ease of mathematical presentation, in the following, we work with continuous time rather than the discretely sampled data and extend the $\mathbf{W}$ factors, $\mathbf{H}$ factors, and data matrix $\mathbf{X}$ through zero-padding on both ends so that:

$$Corr(\overrightarrow{\mathbf{W}}_i^{(n)}, \overrightarrow{\mathbf{X}}^{(n)})(t) = \int_{-\infty}^{\infty} \overrightarrow{\mathbf{W}}_{i\ell}^{(n)} \overrightarrow{\mathbf{X}}_{\ell+t}^{(n)} d\ell \tag{54}$$

and

$$Conv(\overrightarrow{\mathbf{H}}_j, \overrightarrow{\mathbf{s}}) = \int_{-\infty}^{\infty} \overrightarrow{\mathbf{H}}_{j\tau} \overrightarrow{\mathbf{s}}_{t-\tau} d\tau \tag{55}$$

Recall that the Fourier transform is defined as:

$$\hat{f}(\omega) \equiv \mathcal{F}(f(t)) \equiv \int_{-\infty}^{\infty} f(t) e^{-i\omega t} dt \tag{56}$$

with inverse Fourier transform:

$$f(t) = \mathcal{F}^{-1}(\hat{f}(\omega)) \equiv \frac{1}{2\pi} \int_{-\infty}^{\infty} \hat{f}(\omega) e^{+i\omega t} d\omega \tag{57}$$

Now recall some basic features of Fourier transforms of correlation and convolution integrals:

$$\mathcal{F}(Conv(f(t), g(t))) = \hat{f}(\omega)\hat{g}(\omega) \tag{58}$$

$$\mathcal{F}(Corr(f(t), g(t))) = \hat{f}(-\omega)\hat{g}(\omega) \tag{59}$$

$$f(t) \cdot g(t) = \int_{-\infty}^{\infty} f(t)g(t)dt = Corr_{t=0}(f,g) = \mathcal{F}^{-1}\left[\hat{f}(-\omega)\hat{g}(\omega)d\omega\right]_{t=0}$$
$$f(t) \cdot g(t) = \frac{1}{2\pi} \int_{-\infty}^{\infty} \hat{f}(-\omega)\hat{g}(\omega)d\omega \tag{60}$$

This final identity, known as Parseval's theorem, says that the inner product (dot product) between two functions evaluated in the time and frequency domain are equivalent up to a proportionality constant of $1/(2\pi)$. With the above identities, we can calculate our quantity of interest:

$$\mathbf{R}_{ij} = \left[ \sum_{n=1}^{N} Corr(\overrightarrow{\mathbf{W}}_i^{(n)}, \overrightarrow{\mathbf{X}}^{(n)}) \right] \cdot Conv(\overrightarrow{\mathbf{H}}_j, \overrightarrow{\mathbf{s}}) \tag{61}$$

First, define:

$$A(t) = \left[ \sum_{n=1}^{N} Corr(\overrightarrow{\mathbf{W}}_i^{(n)}, \overrightarrow{\mathbf{X}}^{(n)}) \right] \tag{62}$$

$$B(t) = Conv(\overrightarrow{\mathbf{H}}_j, \overrightarrow{\mathbf{s}}) \tag{63}$$

From **Equation 60** (Parseval's theorem):

$$\mathbf{R}_{ij} = \frac{1}{2\pi} \int_{-\infty}^{\infty} \hat{A}(-\omega)\hat{B}(\omega)d\omega \tag{64}$$

Then, from **Equations 58 and 59**, we have:

$$\mathbf{R}_{ij} = \frac{1}{2\pi} \sum_{n=1}^{N} \int_{-\infty}^{\infty} d\omega \hat{\mathbf{W}}_i^{(n)}(\omega)\hat{\mathbf{X}}^{(n)}(-\omega)\hat{\mathbf{H}}_j(\omega)\hat{s}(\omega) \tag{65}$$

The above formula shows that:

1. Viewed in the frequency domain, the x-ortho penalty reduces to a (sum over neurons and frequencies of a) simple product of Fourier transforms of the four matrices involved in the penalty.
2. The smoothing can equally well be applied to $\mathbf{H}$ or $\mathbf{W}$ or $\mathbf{X}$. (For $\mathbf{X}$, note that for symmetric smoothing function $s(t) = s(-t)$, we also have $\hat{s}(\omega) = \hat{s}(-\omega)$.)
3. One can view this operation as either of the below:
    a. First correlate $\mathbf{W}$ and $\mathbf{X}$ by summing correlations of each row, and then calculate the overlap with the smoothed $\mathbf{H}$, as described in the main text: $\mathbf{R}_{ij} = \frac{1}{2\pi} \sum_{n=1}^{N} \int_{-\infty}^{\infty} d\omega \left[ \hat{\mathbf{W}}_i^{(n)}(\omega)\hat{\mathbf{X}}^{(n)}(-\omega) \right] \left[ \hat{\mathbf{H}}_j(\omega)\hat{s}(\omega) \right]$
    b. Correlate $\mathbf{H}$ with each row of $\mathbf{X}$ and then calculate the overlap of this correlation with the corresponding smoothed row of $\mathbf{W}$. Then sum over all rows: $\mathbf{R}_{ij} = \frac{1}{2\pi} \sum_{n=1}^{N} \int_{-\infty}^{\infty} d\omega \left[ \hat{\mathbf{H}}_j(\omega)\hat{\mathbf{X}}^{(n)}(-\omega) \right] \left[ \hat{\mathbf{W}}_i^{(n)}(\omega)\hat{s}(\omega) \right]$

## Fourier representation of the traditional regularization approximation of the x-ortho penalty

We now proceed to show how the x-ortho penalty can be approximated by a traditional (data-free) regularization, expressing the results in the frequency domain. As in the time domain analysis, we consider the approximation in which the data $\mathbf{X}$ are nearly perfectly reconstructed by the convNMF decomposition ($\mathbf{X} \approx \mathbf{W} \circledast \mathbf{H}$).

Noting that this decomposition is a sum over factors of row-by-row ordinary convolutions, we can write the Fourier analog for each row of $\mathbf{X}$ as:

$$\hat{\mathbf{X}}^{(n)}(\omega) \approx \sum_k \hat{\mathbf{W}}_k^{(n)}(\omega)\hat{\mathbf{H}}_k(\omega) \tag{66}$$

Thus, substituting $\mathbf{X}$ with the reconstruction, $\mathbf{W} \circledast \mathbf{H}$ in **Equation 65**, we have:

$$\mathbf{R}_{ij} \approx \frac{1}{2\pi} \sum_k \int_{-\infty}^{\infty} d\omega \left[ \sum_{n=1}^{N} \hat{\mathbf{W}}_i^{(n)}(\omega)\hat{\mathbf{W}}_k^{(n)}(-\omega) \right] \left[ \hat{\mathbf{H}}_k(-\omega)\hat{\mathbf{H}}_j(\omega)\hat{s}(\omega) \right] \tag{67}$$

As in the time domain analysis, we separate the sum over $k$ into three cases: $k = i$, $k = j$, and $k \neq i, j$. Recall that for real numbers, $\hat{f}(-\omega) = \hat{f}^*(\omega)$, and $\hat{f}(\omega)\hat{f}^*(\omega) = |\hat{f}(\omega)|^2$. Thus, separating the sum over $k$ into the three cases, we have:

$$\mathbf{R}_{ij} = \frac{1}{2\pi} \int_{-\infty}^{\infty} d\omega \left[ \sum_{n=1}^{N} \left| \hat{\mathbf{W}}_i^{(n)}(\omega) \right|^2 \right] \left[ \hat{\mathbf{H}}_i(-\omega)\hat{\mathbf{H}}_j(\omega)\hat{s}(\omega) \right] + $$
$$\frac{1}{2\pi} \int_{-\infty}^{\infty} \left| \hat{\mathbf{H}}_j(\omega) \right|^2 \left[ \sum_{n=1}^{N} \hat{\mathbf{W}}_i^{(n)}(\omega)\hat{\mathbf{W}}_j^{(n)}(-\omega)\hat{s}(\omega) \right] + Y \tag{68}$$

where $Y$ represents the remaining terms for which $k \neq i, j$.

We can obtain a more symmetric form of this equation by summing the contributions of factors $i$ and $j$, $\mathbf{R}_{ij} + \mathbf{R}_{ji}$. For symmetric smoothing functions $s(t) = s(-t)$, for which $\hat{s}(\omega) = \hat{s}(-\omega)$, we obtain:

$$\mathbf{R}_{ij} + \mathbf{R}_{ji} = \frac{1}{2\pi}\int_{-\infty}^{\infty} d\omega \left[\sum_{n=1}^{N}\left|\hat{\mathbf{W}}_{i}^{(n)}(\omega)\right|^{2} + \left|\hat{\mathbf{W}}_{j}^{(n)}(\omega)\right|^{2}\right]\left[\hat{\mathbf{H}}_{i}(-\omega)\hat{\mathbf{H}}_{j}(\omega)\hat{\mathbf{s}}(\omega)\right] +$$

$$\frac{1}{2\pi}\int_{-\infty}^{\infty}\left[\left|\hat{\mathbf{H}}_{i}(\omega)\right|^{2} + \left|\hat{\mathbf{H}}_{j}(\omega)\right|^{2}\right]\left[\sum_{n=1}^{N}\hat{\mathbf{W}}_{i}^{(n)}(\omega)\hat{\mathbf{W}}_{j}^{(n)}(-\omega)\hat{\mathbf{s}}(\omega)\right] + Y \qquad (69)$$

As in the time domain analysis, the first two terms above have a simple interpretation in comparison to traditional orthogonality regularizations: The first term resembles a traditional regularization of orthogonality in (smoothed) $\mathbf{H}$, but now weighted frequency-by-frequency by the summed power at that frequency in the corresponding $\mathbf{W}$ factors. For sparse (delta-function-like) sequences, the power in $\mathbf{W}$ at each frequency is a constant and can be taken outside the integral. In this case, the regularization corresponds precisely to orthogonality in (smoothed) H, weighted by the summed power in the corresponding $\mathbf{W}$'s. Likewise, the second term above corresponds to a traditional regularization of orthogonality in (smoothed) $\mathbf{W}$, weighted by the summed power at each component frequency in the corresponding $\mathbf{H}$ factors.

Altogether, we see that these terms represent a Fourier-power weighted extension of (smoothed) traditional orthogonality regularizations in $\mathbf{W}$ and $\mathbf{H}$. This weighting may be beneficial relative to traditional orthogonality penalties, since it makes the regularization focus most heavily on the factors and frequencies that contribute most to the data reconstruction.

Finally, we consider the remaining terms in the cost function, for which $k \neq i,j$. As noted previously, these terms are only relevant when the factorization contains at least three terms, so cannot be seen in the simple Type 1, 2 and 3 cases illustrated in *Figure 1*. These terms have the form:

$$\frac{1}{2\pi}\sum_{k\neq i,j}\int_{-\infty}^{\infty} d\omega \left[\sum_{n=1}^{N}\hat{\mathbf{W}}_{i}^{(n)}(\omega)\hat{\mathbf{W}}_{k}^{(n)}(-\omega)\right]\left[\hat{\mathbf{H}}_{k}(-\omega)\hat{\mathbf{H}}_{j}(\omega)\hat{\mathbf{s}}(\omega)\right] \qquad (70)$$

To understand how this term contributes, we consider each of the expressions in parentheses. The first expression contains each frequency component of the correlation of the $i^{th}$ and $k^{th}$ factors' $\mathbf{W}$'s. The second expression likewise contains each frequency component of the correlation of the $j^{th}$ and $k^{th}$ factors' $\mathbf{H}$'s. Thus, analogous to the time domain analysis, this term of $\mathbf{R}_{ij}$ contributes whenever there is a factor ($\mathbf{W}_{\cdot k\cdot}$, $\mathbf{H}_{k\cdot}$) that overlaps at any frequency with the $i^{th}$ factor's $\mathbf{W}$ component and the $j^{th}$ factor's $\mathbf{H}$ component. In this manner, this three-factor interaction term effectively enforces competition between factors $i$ and $j$ even if they are not correlated themselves, as demonstrated in *Figure 1* of this Appendix.

