## [Decision Letter]

Thank you for sending your article entitled "Unsupervised discovery of temporal sequences in high-dimensional datasets, with applications to neuroscience" for peer review at *eLife*. Your article is being evaluated by Timothy Behrens as the Senior Editor, a Reviewing Editor, and three reviewers.

While the reviewers were generally enthusiastic about the work, major concerns were brought up that raise the question of whether this work is appropriate for the general readership of *eLife*. All reviewers agreed that these major concerns require essential revisions and thus would like to see an action plan that addresses these concerns before they issue a formal decision. In particular, the reviewers were not convinced that this method constitutes a major advance over current state-of-the-art methods in the field. Also, the selection of hyperparameters was not convincingly justified. The reviews are included below in their entirety.

*Reviewer #1:*

This paper introduces a matrix factorization-based method, seqNMF, for extracting temporal sequences from high-dimensional data. The authors convincingly demonstrate that seqNMF performs well in artificial and real datasets. I believe seqNMF will be a useful tool for a wide range of applications. After its appearance at COSYNE, the tool has already created considerable excitement in the computational and systems neuroscience community.

I found the paper to be very well written and the results clearly presented. However, some points need improvement:

1) While Type 1, 2, and 3 errors are clearly explained, I thought better naming could improve the readability of the paper a lot.

2) Building into equation 6, the authors discuss two alternative regularizers, and claim that they would fail at preventing Type 1, 2 and 3 errors simultaneously. While the arguments are clear, an experimental demonstration would be more convincing.

3) In subsection “SeqNMF: A constrained convolutional non-negative matrix factorization”, authors call ||M||_1,i≠j_ a norm on M, which technically isn't correct (Under this definition, a diagonal matrix would have a zero norm).

4) I have trouble with understanding the first method of choosing *λ*. Is this method only applicable to data for which ground-truth is known? Or are the authors suggesting to choose *λ* between 2*λ_0_* and 5*λ_0_* for any dataset? Please clarify what the suggested method is.

*Reviewer #2:*

This paper describes a targeted dimensionality reduction approached, called seqNMF, to identify sequences in neural data. The approach is an improvement on previously published methods. The method will be of interest to its target audience. Also, the method is well described in straightforward terms, and the authors do a great job of describing how to use the method. They provide good examples, both from simulated data and multiple types of real data. They also give a good understanding of how the method can be tuned, such as using the *λ* value, to explore sequences with different levels of granularity or to test different ideas about the data. The paper is well written and is presented in a fair manner. I am therefore supportive of this paper and suggest publication.

*Reviewer #3:*

Overview: This manuscript aims to address the problem of automated discovery of temporal sequences from neural data. The authors report on a modification (seqNMF) of the convolutional non-negative matrix factorization (convNMF) algorithm targeted to address purported issues in the base method. The fundamental issue that seqNMF attempts to address is the redundancy/replication of learned bases by the vanilla convNMF algorithm. The authors use extensive numerical studies to characterize the performance of their algorithm. The target biological datasets are neural activity (spike trains and calcium signals) from hippocampus and nucleus HVC of the zebra finch. This paper is generally well written, though it could be streamlined substantially to accelerate the reader through the manuscript. However, for reasons elaborated below, it is unclear how much of an advance the reported method is over the current state of the art in the field. Most importantly, the authors seem unaware of recent advances in the statistical-machine learning literature that aim at addressing the issue of non-redundant bases learning by NMF algorithms. Furthermore, the heuristic used for selection of the penalty hyperparameter is not adequately justified. As such, the impact/novelty of the proposed algorithm has not been evaluated or demonstrated: quantitative comparisons to the state-of-the-art is required for evaluation of proposed new analytic methods. While these issues could potentially be addressed, in my opinion, this work does not present a clear major improvement over existing methods and thus belongs in a much more specialized venue.

Major Critiques:

1) Insufficient comparison to state-of-the-art methods. The central issue that the seqNMF algorithm attempts to solve is "To reduce the occurrence of redundant factors in convNMF […]". The authors identify three types of bases learning consistency errors. To address these errors, the authors augment an existing penalty on the reconstruction cost function to encourage uncorrelated bases and loadings. This is a reasonable strategy but is by no means the only approach. Indeed, as mentioned by the authors:

-Direct selection of the rank K vs. hyperparameter strength (*λ*)

-Use of a sparsity penalty (e.g., L1-regularization)

While these approaches are mentioned, and sometimes examined, the benefit of the proposed method over these approaches has not been demonstrated in either synthetic or neurobiological data. For example, if one were to attempt to optimize the number of bases (K) and the regularization strength (*λ*) in a sparse convNMF, how do the performance measures on the synthetic data sets with various noise levels stack up? Note that the results of Figure 2C,D in this regard are not at all convincing, as the rank K of the factorization of convNMF is not optimized: indeed, I find this figure to be misleading, as it is not an 'apples-to-apples' comparison (also, I believe there is a mislabeling of convNMF vs. seqNMF in this figure).

More seriously, the authors seem unaware of recent advances in the statistical-machine learning community which directly address the issue of learning non-redundant bases. For example, the 'stability NMF' algorithm (Wu et al., 2016) was recently successfully applied to extract biologically meaningful patterns from diverse data sets. More recently, the UoI-NMFcluster algorithm (Ubaru et al., 2018) has been shown to extract the precise generative bases, and assign sparse reconstruction weights to those bases, from noisy synthetic and neurobiological data. Both of these methods aim to select bases that are minimally correlated with each other. While these methods have not been applied to sequences per se, there is no conceptual reason why they could not be. For example, if one allows for a sliding cross-correlation, it is likely that both stability NMF and UoI-NMF could deal with temporal jitter. Can these methods be modified to provide even greater robustness for learning non-redundant bases for sequences?

In summary, the authors have not demonstrated the broad and robust improvement of the proposed algorithm relative to other methods in the field.

2) Method for selecting regularization parameters. In the seqNMF algorithm, the balance between reconstruction and un-correlated bases is modulated by the *λ* hyperparamter. The authors provide various heuristics for choosing *λ* to find the correct bases. The main heuristic used for the main figure use a heuristic of 2-5x of the *λ* value that gives the cross-over in normalized reconstruction vs. correlation costs (*λ_0_*). The authors justify this as giving good results in the synthetic data sets and having a reasonable overlap with cross-validated reconstruction error. However, for many of the synthetic and real data sets, lambda0 also lies at the most sensitive part of the range, and small modulations lead to large changes in the trade-off of reconstruction vs. correlation. That is, the results are likely to be very sensitive to this selection. Furthermore, the selected values often result in very poor data reconstruction (e.g., Figure 6C,D and Figure 7B,F). As quantified, this is a big problem for 'interpretability', as it is very hard to interpret the results of a method that does not have good data prediction quality. In science, ground-truth is not known, and a major metric we have to quantitatively evaluate methods is reconstruction error/parsimony. If the authors could show that, something like cross-validated Bayesian Information Criterion, was optimized for the selected values of *λ*, that would be much more convincing.

In summary, the heuristic method used to select the regularization strength, which is a major component of the reported work, is not sufficiently justified from a statistical-machine learning perspective.

---

## [Author Response]

[Editors' note: the authors’ plan for revisions was approved and the authors made a formal revised submission.]

While the reviewers were generally enthusiastic about the work, major concerns were brought up that raise the question of whether this work is appropriate for the general readership of eLife. All reviewers agreed that these major concerns require essential revisions and thus would like to see an action plan that addresses these concerns before they issue a formal decision. In particular, the reviewers were not convinced that this method constitutes a major advance over current state-of-the-art methods in the field. Also, the selection of hyperparameters was not convincingly justified. The reviews are included below in their entirety.

We thank the reviewers for their thorough and insightful reading of our manuscript. We believe their criticisms will lead to a substantial improvement in our work. Here we address these criticisms and outline improvements to the paper.

We would like to emphasize that the aim of this paper is to provide the neuroscience community with a broadly useful and largely automated method for identifying temporal patterns (which we refer to as “sequences”) in neural data. Such methods do not currently exist.

Thus, we think that, for the neuroscience community, seqNMF is a major advance and fills an important gap. Sequential firing patterns have appeared in a large range of important studies in different brain regions and different behaviors (Hahnloser, 2002; Harvey, 2012; Okubo, 2015; Fujisawa, 2008; Pastalkova, 2008; MacDonald, 2011). These studies are all from top neuroscience labs with strong quantitative expertise, and yet in each of these studies, the sequences were extracted manually by alignment of neural activity with external behavioral or task events. This suggests a profound lack of existing methods in the field for automatic extraction of sequential structure in neuronal data.

Towards our goal of making our method practical and easy-to-adopt, we have intentionally worked to build upon core existing approaches from the machine learning community. We have attempted to write the paper in an accessible and pedagogical format so that the broader neuroscience community can easily apply the methods described, using the easy-to-use code we have posted publicly.

Our goal has therefore been to bring together a variety of statistical approaches: Convolutional matrix factorization, regularization strategies to encourage orthogonal factors, cross-validation, and now (thanks to reviewer 3) stability-based measures for determining model hyperparameters. Many of these individual concepts can be found by digging through different pieces of previous literature; however, repurposing and refining models for practical applications has a strong history in machine learning and computational neuroscience and can constitute a significant advance for the field. To make this clearer in our current version of the manuscript we have used seqNMF only to refer to a toolbox containing all of our contributions. The penalty term previously referred to as seqNMF is not referred to as the cross-orthogonality (x-ortho) penalty term. Thus, even if our paper did not represent a technical advance in statistical methodology, our careful analysis of the performance of regularized convNMF on different types of synthetic and real neural (and behavioral) data represents a significant advance of interest to the broad readership of *eLife*.

That said, we still take very seriously the concerns of Reviewer 3 about the novelty of our method. We thank the reviewer for pointing out the richness of recent advances in the area of preventing bases learning consistency errors, some of which we were unaware of. Thanks to the suggestion of the reviewer we have now incorporated some of these recent advances in our method.

In short, we would prefer to think of our contribution not as a detailed benchmarking of different methods of regularizing convolutional NMF, but rather as a toolkit for neuroscientists (who may not be statistics experts) to automatically and reliably extract sequences from neural data, to test their statistical significance, and to flexibly fine-tune the structure of the factorization. Finally, we have endeavored to present these ideas within a pedagogical framework so that readers can readily understand, and potentially build on the method.

Reviewer #1:This paper introduces a matrix factorization-based method, seqNMF, for extracting temporal sequences from high-dimensional data. The authors convincingly demonstrate that seqNMF performs well in artificial and real datasets. I believe seqNMF will be a useful tool for a wide range of applications. After its appearance at COSYNE, the tool has already created considerable excitement in the computational and systems neuroscience community.I found the paper to be very well written and the results clearly presented. However, some points need improvement:1) While Type 1, 2, and 3 errors are clearly explained, I thought better naming could improve the readability of the paper a lot.

We attempted to change the error types to: duplication, fragmentation and duplication with independence but felt that this made the paper more difficult to read. Perhaps there are other better names which would improve readability however we have chosen to keep Type 1, 2 and 3 for the time being.

2) Building into equation 6, the authors discuss two alternative regularizers, and claim that they would fail at preventing Type 1, 2 and 3 errors simultaneously. While the arguments are clear, an experimental demonstration would be more convincing.

We have added a supplementary figure (Figure 1—figure supplement 1) illustrating each type of error on reconstructions of synthetic data. We have also quantified the rates of occurrence of each error type for the different regularizers described in the paper. This analysis illustrates the pattern described in the text.

In addition, we have added an appendix analyzing how the penalty used in seqNMF relates to traditional regularizers.

3) In subsection “SeqNMF: A constrained convolutional non-negative matrix factorization”, authors call ||M||_1,i≠_j a norm on M, which technically isn't correct (Under this definition, a diagonal matrix would have a zero norm).

We changed the text to refer to this as a seminorm.

4) I have trouble with understanding the first method of choosing λ. Is this method only applicable to data for which ground-truth is known? Or are the authors suggesting to choose λ between 2 λ_0_ and 5λ_0_ for any dataset? Please clarify what the suggested method is.

The reviewer’s interpretation of our method is correct; we are suggesting that the choice of *λ* between 2*λ_0_* and 5*λ_0_* is a good starting point for any dataset. This range is based on the performance of the algorithm on synthetic datasets for which the ground truth is known.

Further clarification: The procedure for choosing *λ* is motivated by the common-sense observation that *λ* provides a way to weight the priority given to reconstruction error vs redundancy (correlation cost). In the datasets we tested, values between 2*λ_0_* and 5*λ_0_* tended to yield reasonable results, reducing redundancy without sacrificing too much reconstruction error. In certain applications, the desired tradeoff may be different (e.g., when the main priority is reconstruction error, and some redundancy is tolerable), in which case the intuition behind our procedure may be useful for selecting *λ* (e.g., choose lower *λ*). We have made this intuition clearer in the text, so that future users of our algorithm will have a better idea of the tradeoffs being made when choosing different values of *λ*. That being said, based on our synthetic data and examining the results with different values for the real data, we suggest starting with a λ between 2 and 5 for any data set. In the cases we have considered where the ground-truth is known, we have shown that this strategy produces results that agree with the ground-truth (Figure 4). Furthermore, these results were relatively insensitive to the precise choice of *λ*. That is, ground-truth factors could be recovered across a wide range of *λ* (Figures 4 and Figure 2—figure supplement 2; also compare Figure 3 with Figure 3—figure supplement 1).

Reviewer #3:Overview: This manuscript aims to address the problem of automated discovery of temporal sequences from neural data. The authors report on a modification (seqNMF) of the convolutional non-negative matrix factorization (convNMF) algorithm targeted to address purported issues in the base method. The fundamental issue that seqNMF attempts to address is the redundancy/replication of learned bases by the vanilla convNMF algorithm. The authors use extensive numerical studies to characterize the performance of their algorithm. The target biological datasets are neural activity (spike trains and calcium signals) from hippocampus and nucleus HVC of the zebra finch. This paper is generally well written, though it could be streamlined substantially to accelerate the reader through the manuscript. However, for reasons elaborated below, it is unclear how much of an advance the reported method is over the current state of the art in the field. Most importantly, the authors seem unaware of recent advances in the statistical-machine learning literature that aim at addressing the issue of non-redundant bases learning by NMF algorithms. Furthermore, the heuristic used for selection of the penalty hyperparameter is not adequately justified. As such, the impact/novelty of the proposed algorithm has not been evaluated or demonstrated: quantitative comparisons to the state-of-the-art is required for evaluation of proposed new analytic methods. While these issues could potentially be addressed, in my opinion, this work does not present a clear major improvement over existing methods and thus belongs in a much more specialized venue.

We thank the reviewer for a thorough reading and thoughtful response to our manuscript.

The central aim of our paper is to present a simple and widely applicable toolset for neuroscientists to detect sequences in their data. Reducing the occurrence of redundant factors in convNMF was an important part of this aim, but not an endpoint in itself. Thus, we looked to the statistical-machine learning literature for inspiration, not for competition. (Indeed, we thank the reviewer for pointing out advances in this field that have turned out to be very effective.) We purposely targeted *eLife*, rather than a computer science or machine learning venue, because the appropriate audience for this work is the neuroscience community, which has a long-standing interest in neural sequences from both experimental and theoretical perspectives (e.g. Hahnloser, 2002; Harvey, 2012; Okubo, 2015; Fujisawa, 2008; Pastalkova, 2008; MacDonald, 2011; Brody, 1999; Mokeichev, 2007; Schrader, 2008; Gerstein, 2012; Torre, 2016; Russo, 2017’ Grossberger, 2018; Quaglio, 2018).

Thus, our contribution is less about a particular advance in constrained optimization, but rather about adapting and applying a collection of statistical tools to the problem of discovering interpretable sequential structure in neuroscience data. These include: the use of convolutional NMF on a variety of neural and behavioral datasets, constraints to reduce redundant factorizations, methods to select hyperparameters, estimation of statistical significance, quantification of robustness to different common noise types, and regularization to shape the structure of factorizations (i.e. parts vs events).

Despite the interest in neural sequences, factorization-based approaches have been underutilized in the field. This is likely because the appropriate set of statistical tools had not yet been assembled, adapted and applied to neuroscience data in a convincing manner. We believe that our work represents such an advance.

Major Critiques:1) Insufficient comparison to state-of-the-art methods. The central issue that the seqNMF algorithm attempts to solve is "To reduce the occurrence of redundant factors in convNMF […]". The authors identify three types of bases learning consistency errors. To address these errors, the authors augment an existing penalty on the reconstruction cost function to encourage uncorrelated bases and loadings. This is a reasonable strategy but is by no means the only approach. Indeed, as mentioned by the authors:-Direct selection of the rank K vs. hyperparameter strength (λ)-Use of a sparsity penalty (e.g., L1-regularization)While these approaches are mentioned, and sometimes examined, the benefit of the proposed method over these approaches has not been demonstrated in either synthetic or neurobiological data. For example, if one were to attempt to optimize the number of bases (K) and the regularization strength (λ) in a sparse convNMF, how do the performance measures on the synthetic data sets with various noise levels stack up?

We agree with the reviewer that it would be useful to compare our approach to the others mentioned (i.e. direct selection of rank K, or the use of a sparsity penalty) and we have added additional results exploring these approaches.

Direct selection of K

We had previously shown that direct selection of K can sometimes be carried out using cross-validated error performance on held-out data. However, following the reviewer’s suggestion of extending stability NMF (and the *diss* metric, see below) to the convolutional case has yielded significantly better performance for directly estimating the rank K. We have therefore added a new main figure (Figure 5) and a new supplementary figure (Figure 5—figure supplement 1) to the paper illustrating the stability NMF approach and have modified the text to present this as an additional, complementary method for minimizing bases consistency errors.

L1-sparsity

One of the advantages of the cross orthogonality penalty is that it consists of only a single term to penalize correlations between different factors, and thus requires the determination of only a single hyperparameter (*λ*). This contrasts with a different potential approach to minimize redundant factorizations, namely incorporating a sparsity constraint on W and H of the form *λ_w_*||W||_1_+ *λ_h_*||H||_1_. We first wanted to understand the dependence of the sparsity approach on the hyperparameters (*λ_h_* and *λ_w_*). Performance was quantified by measuring both the similarity to ground-truth and the probability of obtaining the correct number of significant factors using synthetic data sets with intermediate levels of each of the four noise types. We found that the similarity to ground-truth and probability of obtaining the correct number of significant factors varies in a complex manner as a function of the two *λ* parameters (Figure 4—figure supplement 4).

We next wondered how the ‘optimal’ performance of the sparsity approach (at the best choice of *λ_h_* and *λ_w_*) compares with the ‘optimal’ performance of the x-ortho approach (at the best choice of the cross orthogonality *λ*). [See below for more information on how this was done.] Having chosen the best *λ* for L1-sparsity and cross orthogonality for each noise condition, we ran the factorization for each method and quantified the distribution of performance (again using similarity and the number of significant factors). We found that L1 sparsity constraint yielded an overall performance quite similar to the cross orthogonality regularizer (Figure 4—figure supplement 5). Interestingly though, there are some consistent differences in the performance of these two constraints depending on the noise type: the x-ortho approach performs slightly better with warping and participation noise, while L1 sparsity performs slightly better with jitter and additive noise (Figure 4—figure supplement 5).

Given these findings, we feel that the x-ortho penalty represents a practical advance over the sparsity approach. The two methods perform similarly once the optimal values of the hyperparameters is determined, but in cases where the ground truth is not known, it will be simpler to find the optimal value of a single parameter (by using the seqNMF method for selecting *λ*), than two parameters for sparsity. Of course, it may be also possible to develop a method to find suitable values of *λ_h_* and *λ_w_* for the sparsity approach, but given the strong performance of the x-ortho penalty term, there is little motivation to develop such a method.

More detailed methods for L1 sparsity comparison

Optimal values of *λ_h_* and *λ_w_* for the sparsity method were selected by computing the composite performance score at each value of *λ_h_* and *λ_w_* for each noise type (see arrow in Figure 4—figure supplement 4A). The composite performance is the same as that used in Figure 4F, namely the element-wise product of the similarity to ground-truth and the fraction of fits with the correct number of factors. The peak value of this composite performance score yielded a pair of lambdas at which the sparsity penalties gave the best performance on this dataset. For the comparison, the same procedure was used to select the optimal *λ* for the x-ortho penalty in each noise condition. The performance of sparsity and x-ortho were each evaluated at their optimal lambdas by fitting the data 100 times from random initial conditions.

Note that the results of Figure 2C,D in this regard are not at all convincing, as the rank K of the factorization of convNMF is not optimized: indeed, I find this figure to be misleading, as it is not an 'apples-to-apples' comparison (also, I believe there is a mislabeling of convNMF vs. seqNMF in this figure).

With regard to Figure 2, we apologize that our intention was not clear. Our intention was simply to illustrate the effect of the regularizer, rather than to formally compare to convNMF as a competing method. This is now clarified in the text.

More seriously, the authors seem unaware of recent advances in the statistical-machine learning community which directly address the issue of learning non-redundant bases. For example, the 'stability NMF' algorithm (Wu et al., 2016) was recently successfully applied to extract biologically meaningful patterns from diverse data sets. More recently, the UoI-NMFcluster algorithm (Ubaru et al., 2018) has been shown to extract the precise generative bases, and assign sparse reconstruction weights to those bases, from noisy synthetic and neurobiological data. Both of these methods aim to select bases that are minimally correlated with each other. While these methods have not been applied to sequences per se, there is no conceptual reason why they could not be. For example, if one allows for a sliding cross-correlation, it is likely that both stability NMF and UoI-NMF could deal with temporal jitter. Can these methods be modified to provide even greater robustness for learning non-redundant bases for sequences?In summary, the authors have not demonstrated the broad and robust improvement of the proposed algorithm relative to other methods in the field.

We thank the reviewer for these suggestions for alternative possible methods to constrain convNMF factorizations, and to reduce bases learning consistency errors. Indeed, we were not aware of these recent exciting advances in the field and have explored the application of both stability NMF and UoI to convNMF.

Stability NMF

We have successfully adapted stability NMF (Wu et al., 2016) using the dissimilarity (diss) metric to account for temporally-shifted correlations. As the reviewer pointed out, different runs of convNMF may produce factors which are shifted in time (in both H and W) and thus have artificially low correlations, confounding the stability measure described in Wu et al. In our initial exploration of this metric, we observed that smoothing the factors worked sometimes but found that a much more robust way of overcoming this misalignment problem is to apply the stability analysis to the reconstructions of each individual factor (that is, w_k_⊛ h_k_). Because this is a more reliable method of optimizing K for convNMF than the cross-validation approach we developed previously, we have added a new main figure (Figure 5) and a new supplementary figure (Figure 5—figure supplement 1) and have incorporated this into our Results section “Strategy for choosing K rather than choosing *λ*?”

Union of Intersections

We next wondered if the approach of UoI-NMF could be adapted to improve the performance of seqNMF. To apply this method to the convolutional case we first had to make several adjustments to the UoI algorithm. First, we simplified things by skipping the first step of estimating the appropriate rank using stability NMF and simply chose K to be the ground-truth rank of our synthetic data. Because convolutional NMF relies on structure across observations in time it is not possible to bootstrap our data in the conventional way by taking random samples of observations. Instead we took the approach that we used for cross-validation in Figure 5—figure supplement 2; we randomly held out individual bins from the dataset during fitting and defined the remaining data as our bootstrapped sample. Finally, rather than solving a non-negative least squares problem, we took advantage of the multiplicative update rules to constrain our factorizations. After taking the intersections of our factors from many different bootstrapped fits we defined an intersection mask where an element was defined as 1 if it was a member of the intersection and zero otherwise. We then used this mask as an initialization of our factorization, and the multiplicative nature of the updates ensures that elements which are initialized as 0 remain fixed at zero throughout the optimization thus enforcing the intersection.

Before applying our implementation of UoI-NMF to convNMF, we first tested that our changes to the UoI algorithm did not negatively impact its ability to robustly identify bases in an NMF model (Author response image 1). We generated a ground-truth dataset using synchronous patterns and compared the ability of UoI-NMF and NMF to extract the ground-truth bases. As expected, our adapted implementation of UoI-NMF extracted factors that were much more similar to the ground-truth set than NMF for a range of noise conditions (Author response image 1).

**Author response image 1. respfig1:** Validation of our adaptations of UoI-NMF. Similarity to ground-truth as a function of additive noise level for UoI-NMF factorizations and NMF factorizations using the algorithmic changes detailed above. Note that UoI-NMF extracts bases which are closer to the ground-truth patterns than regular NMF.

After ensuring that our adaptations of the UoI-NMF algorithm still yielded good results in the NMF case, we tested it on convNMF, but found that it gave mixed results. In the best cases, UoI-convNMF returned factors which were a 5-10% better match to the ground-truth factors than regular convNMF, but in some conditions, UoI-convNMF returned factors which were a 5-10% worse match, and our attempts to further adapt the algorithm or change parameters sometimes yielded even worse results. We believe that this is primarily due to the challenge that convNMF factors on individual runs can be shifted in time relative to one another, as mentioned above. This could, in principle, cause problems during the intersection step of the algorithm, because misaligned bases will tend not to overlap. In summary, the UoI approach is a promising way to address bases correlations in convNMF, but we believe that the additional extensions necessary for good performance in the convNMF model are beyond the scope of our paper. In our Discussion section, we have added a description of the UoI-NMF method, highlighting it as a promising possible extension for discovering robust and sparse sequential bases.

2) Method for selecting regularization parameters. In the seqNMF algorithm, the balance between reconstruction and un-correlated bases is modulated by the λ hyperparamter. The authors provide various heuristics for choosing λ to find the correct bases. The main heuristic used for the main figure use a heuristic of 2-5x of the λ value that gives the cross-over in normalized reconstruction vs. correlation costs (λ_0_). The authors justify this as giving good results in the synthetic data sets and having a reasonable overlap with cross-validated reconstruction error. However, for many of the synthetic and real data sets, lambda0 also lies at the most sensitive part of the range, and small modulations lead to large changes in the trade-off of reconstruction vs. correlation. That is, the results are likely to be very sensitive to this selection. Furthermore, the selected values often result in very poor data reconstruction (e.g., Figure 6C,D and Figure 7B,F). As quantified, this is a big problem for 'interpretability', as it is very hard to interpret the results of a method that does not have good data prediction quality. In science, ground-truth is not known, and a major metric we have to quantitatively evaluate methods is reconstruction error/parsimony. If the authors could show that, something like cross-validated Bayesian Information Criterion, was optimized for the selected values of λ, that would be much more convincing.In summary, the heuristic method used to select the regularization strength, which is a major component of the reported work, is not sufficiently justified from a statistical-machine learning perspective.

We agree that our method for choosing *λ* is heuristic and is not well motivated formally from a machine learning perspective. However, we have also confirmed that our heuristic method agrees with the results of a cross-validation procedure, as presented in Figure 5—figure supplement 2. The reason we kept the heuristic in the main text is because it is more robust in low-noise cases (where overfitting is minimal) and is more interpretable for some noise types such as warping. In our action plan we stated that we would explore the use of cross-validated reconstruction error in the selection of *λ*, but we were unable to find any methods that performed better than our heuristic. Furthermore, our analysis of synthetic datasets reveals that the performance on a given dataset is not very sensitive to the choice of *λ* in the range we have described. In fact, in most cases, we recover the correct number of factors over a range of *λ* spanning nearly an order of magnitude. We have quantified this in a new figure (Figure 4—figure supplement 1).

The reviewer suggests that the optimal value of *λ* often results in very poor data reconstruction. This interpretation of Figure 6C,D and 7B,F, is incorrect. The plots of reconstruction that the reviewer refers to are normalized to go between 0 and 1, and the overall reconstruction error cannot be interpreted from these plots. At low *λ*, reconstruction error reaches a minimum which corresponds to the unregularized convNMF reconstruction error. In contrast, at high *λ*, reconstruction error saturates at a value that is determined by the amount of variance which can be explained by a single factor (often quite a lot). This is because the x-ortho penalty does not affect K=1 factorizations. For the case of single sequential pattern embedded in high noise, the low *λ* region reflects massive over-fitting of noise, whereas the saturated region at high lambdas will represent ideal performance (even though it is the “worst” reconstruction error of any range of *λ* values). For the case of multiple sequences embedded in noise, the leftmost region still represents massive over-fitting while the right-most region represents the error incurred by incorrectly using a 1 factor model. Thus, by definition the best parameter value must lie on the steep region of the reconstruction error as factorizations move from an essentially un-penalized 20 factor factorization to an un-penalized 1 factor factorization. In other words, reconstruction error is large because the data are noisy, and seqNMF is fitting only the repeatable sequences, and not the noise. Furthermore, values of *λ* that produce better reconstruction error do so by overfitting the data, that is, producing factors with lower similarity to ground-truth (Figure 4E and 4K).